# A unified framework for analysis of individual-based models in ecology and beyond

Stephen J. Cornell[1,6]*, Yevhen F. Suprunenko [1,2,6], Dmitri Finkelshtein [3], Panu Somervuo [4] & Otso Ovaskainen [4,5]

Individual-based models, 'IBMs', describe naturally the dynamics of interacting organisms or social or financial agents. They are considered too complex for mathematical analysis, but computer simulations of them cannot give the general insights required. Here, we resolve this problem with a general mathematical framework for IBMs containing interactions of an unlimited level of complexity, and derive equations that reliably approximate the effects of space and stochasticity. We provide software, specified in an accessible and intuitive graphical way, so any researcher can obtain analytical and simulation results for any particular IBM without algebraic manipulation. We illustrate the framework with examples from movement ecology, conservation biology, and evolutionary ecology. This framework will provide unprecedented insights into a hitherto intractable panoply of complex models across many scientific fields.

[1] Institute of Integrative Biology, University of Liverpool, Liverpool L69 7ZB, UK. [2] Department of Plant Sciences, University of Cambridge, Downing Street, Cambridge CB2 3EA, UK. [3] Department of Mathematics, Swansea University, Fabian Way, Swansea SA1 8EN, UK. [4] Organismal and Evolutionary Biology Research Programme, University of Helsinki, Helsinki P.O. Box 65, FI-00014, Finland. [5] Centre for Biodiversity Dynamics, Department of Biology, Norwegian University of Science and Technology, N-7491 Trondheim, Norway. [6] These authors contributed equally: Stephen J. Cornell, Yevhen F. Suprunenko. *email: stephen.cornell@liverpool.ac.uk

There are many systems in biology[1], the social sciences[2] and other disciplines comprising populations of individuals that move and interact with each other, but where the properties of focal interest arise at the whole-population level. In ecology and evolution[1,3,4], for instance, large-scale patterns, such as coexistence, biodiversity, and the evolution of novel types emerge from individual-level processes, such as births, deaths and inter-agent interactions. Historically, however, many fundamental ideas in ecology were developed using differential equation models that treat populations as continuous numbers.

The mathematical transparency of these models means that we understand their predictions completely and can propose general principles, e.g., regarding the number of resources needed by competing species[5] or the processes that destabilise host–parasite dynamics[6]. However, it has long been recognised that these simple models are inadequate for describing many biological phenomena[7]. For example, the conditions for competitors to coexist, and for predator–prey cycles to occur, are altered when space and individual discreteness are considered[8]. Space and stochasticity can similarly play an important role in the dynamics of chemical species [9]. Individual-based (or agent-) based models[10] faithfully capture the discrete and spatial nature of population dynamics, but these are usually studied by computer simulation[1], which only tells us about a limited set of parameter values and not the general model behaviour. They therefore do not yield the sort of general understanding that would be given by mathematical analysis, which encapsulates behaviour across all possible parameter combinations[8].

To use individual-based models to develop general principles, of the sort derived from classic differential equation models, requires a method for analysing them mathematically. Individual-based models can be formulated by spatiotemporal point processes[11], where individuals (or, more generally, entity types, such as juveniles or infected individuals, see Fig. 1a) are created, destroyed and move at rates that can depend on the positions of other individuals in the system, see Fig. 1b. In principle, the dynamics of such systems are described exactly by equations for the time evolution of spatial moments, representing mean density of individuals (first-order moment), spatial covariance (second-order moment) and so on (Fig. 1c). However, in practice there are two obstacles to using these to predict model behaviour. First, they need to be laboriously derived separately for each model, and for each order of moment[12]. Second, the moment equations form an unclosed hierarchy, with the dynamics of each moment depending on the higher order moments, so if the equations for all orders of moment were known it would still not be possible to solve them—even numerically[13]. 'Moment closure' is a widely used approximation scheme that closes the hierarchy by an ad-hoc assumption relating moments of different order[12], but this is an uncontrolled approximation that is not guaranteed to perform well in any particular limit[8,14]. A more reliable alternative based on a perturbation expansion has been proposed, which gives asymptotically exact results when agents interact over large enough scales, but the algebraic burden for this method remains prohibitive because it requires arduous derivations for each particular model[11,13] (Fig. 1d).

Here, we overcome these difficulties by formulating a unified theoretical framework for a wide class of systems, which allows us to derive general analytical results. The results for any particular model within this general class can be obtained directly without further analysis. We make analytical results available to the non-specialist by providing software, which generates mathematical expressions describing a model that the user specifies in an accessible and intuitive way. We first describe our framework and the mathematical results leading to it, and then give three applications that illustrate how our method allows us to answer questions that are not addressable by a simulation approach alone.

## Results

**The framework**. We begin by classifying the participants in demographic processes into three types of individuals (borrowing terminology from chemistry): (i) reactants (that are destroyed by the process); (ii) products (that are created by the process) and (iii) catalysts (that are unaffected by the process, but whose presence affects the rate at which it occurs). For example: a death event has a single reactant and no products or catalysts; dioecious birth has two catalysts (the mother and the father) and one product (the offspring) and movement can be represented by a reactant at the initial position and a product at the final position. This representation can describe processes with an arbitrarily high degree of complexity: the number of reactants, catalysts and products that can participate in any event is unlimited, as is the number of entity types within the system. We could, for instance, model a population where individuals can consume a food item, thereby increasing their energy level, provided they are in the shelter of a tree and that there is another individual helping to capture the prey. In this case, there would be two catalysts (tree and helper), two reactants (original individual and food item) and one product (an individual with a higher energy level than the original), and different entity types would be used to represent trees, food items and individuals of the focal species with different energy levels. This general classification allows us to derive our first main result: an exact expression for the moment equations to all orders for the general model containing processes with arbitrary sets of reactants, products and catalysts and interactions between them (see the Methods section Eq. (3)).

Then, using this exact expression for the general model, we apply a perturbation scheme[13,15] to derive a robust approximation for the dynamics of this general class of model. If the interactions between individuals take place over very large spatial scales, the demographic rates depend only on globally averaged population densities. In that case, the moment equations reduce to 'mean-field' equations—ordinary differential equations of the type used in classical population dynamics. If the interactions are more local, then stochastic and spatial fluctuations cause the model to deviate from the mean-field behaviour. We assume that spatial interactions between individuals depend on their separation $x$ (in $d$-dimensional space) according to interaction 'kernels' of the form $\epsilon^d f(\epsilon x)$, where $1/\epsilon$ is the typical length scale of interactions, and prove mathematically (Supplementary Note 1) that the mean densities and spatial covariance (including autocovariance) satisfy an expansion

$$\text{density} = q + \epsilon^d p + o(\epsilon^d),$$
$$\text{spatial covariance} = \epsilon^d g(\epsilon x) + o(\epsilon^d),$$

where $q$ is the density computed by the mean-field model, $p$ is the correction to the mean-field density due to spatial stochastic fluctuations and $g$ is the dominant contribution to the spatial covariance that describes the degree to which individuals are aggregated or segregated in space ($o(\epsilon^d)$ denotes a term that, when divided by $\epsilon^d$, vanishes when $\epsilon \to 0$). For simplicity, in the above expression we have assumed translational invariance so that the density does not depend on position and the spatial covariance depends only on spatial separation, but our underlying mathematical framework does not have this limitation, see Fig. 1e and Eq. (1) therein. Depending on the scientific question being asked, it may be enough to study the mean-field behaviour described by $q$ alone, whereas $p$ or $g$ will need to be calculated if we are interested in the nature and consequences of spatial patterns that emerge from local interactions. This perturbation

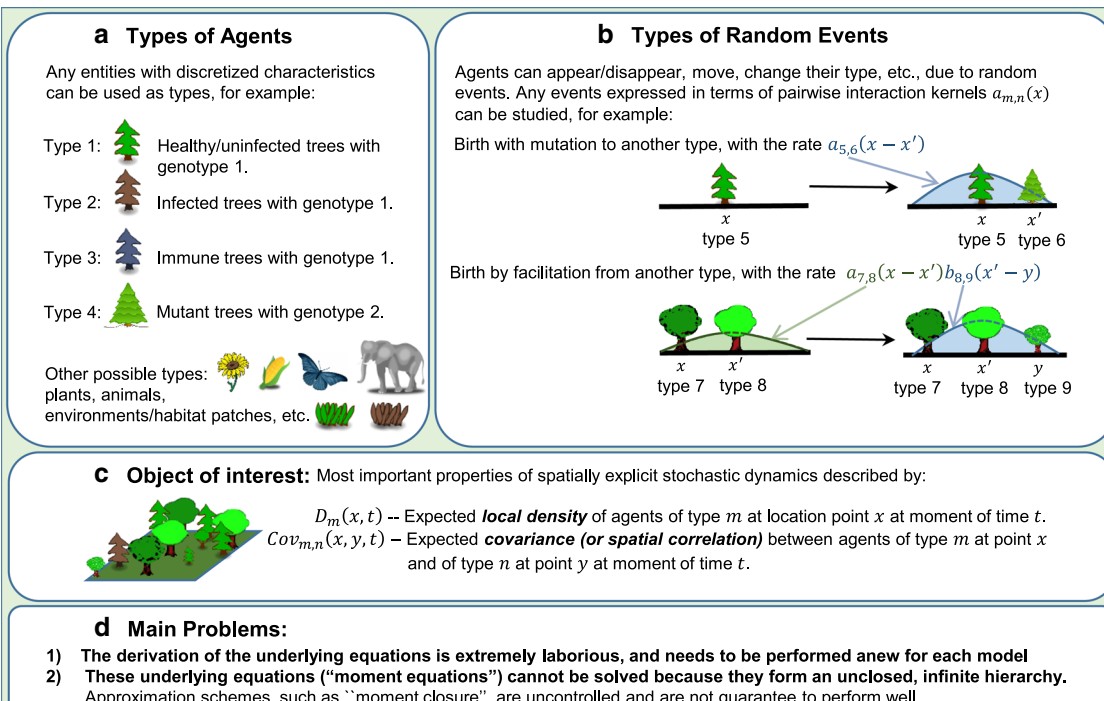

**Fig. 1** Agent-based models in ecology and evolution: the problem, and our solution. Agent-based models consist of entities of different types (panel **a**) that can interact in space in many ways (panel **b**). At any time, the state of the model can be quantified by spatial moments, such as density or spatial correlations (panel **c**). Agent-based models are generally considered too complicated to analyse mathematically (panel **d**). We address this problem by defining a general class of models and processes, for which we provide computer code that yields analytical solutions and runs stochastic simulations (panel **e**)

scheme has been applied to some specific ecological models before, and has been found to perform as well as or better than alternative moment closure approaches[13,15]. The framework described here allows us to apply this technique to a much wider class of models.

Our second main result is an expression for differential equations for $q$, $p$ and $g$ for the general reactant–catalyst–product process (Fig. 1e and Eq. (2) therein, and Eqs. (4), (5), (6) in the Methods section). This allows us to construct directly the perturbation expansion for any model containing any number of processes in our general reactant–catalyst–product class because, even though the interactions between individuals may be nonlinear, the differential equations for $q$, $p$ and $g$ contain

sums of independent contributions from each process. Our results make mathematical analysis available for systems that would be prohibitively complex using other approaches, and, while Eqs. (4), (5), (6) may appear daunting, we have automated this process by writing Mathematica code that computes analytical expressions for general reactant–catalyst–product models (see Supplementary Note 2). To complement and verify the analytical results, we have also written computer code in C that simulates a very broad class of reactant–catalyst–product models (see Supplementary Note 2). This provides a unified framework for analysis and simulation (illustrated in Fig. 2), where the user wanting to study a particular model has only to convert the verbal model description into a graphical model

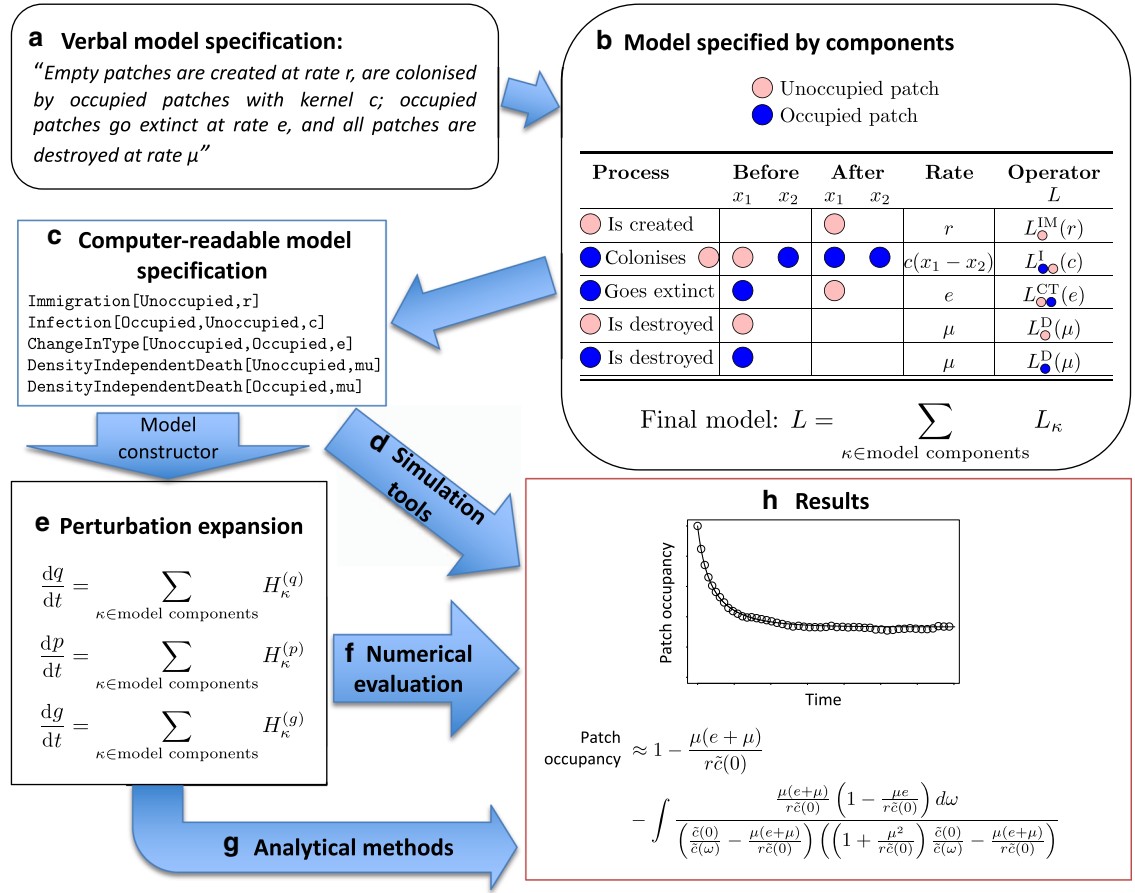

**Fig. 2** Steps in our framework for constructing models and calculating their predictions. These are illustrated using our dynamic landscape metapopulation case study. The user wanting to study a particular model has only to convert the verbal model description (**a**) into a graphical model description (**b**), formulated in terms of a list of possible events and the rates at which they occur. An unambiguous mathematical statement of the model as a spatiotemporal point process is obtained from the operators $L$ for these components[11]; the superscripts denote the corresponding class of reactant–catalyst–product process, namely immigration (IM), infection (I), change of type (CT), density-independent death (D). The colour denotes the type of individual (blue: occupied patch; pink: unoccupied patch). A computer-readable statement of the components (**c**) can be used to generate code for simulating the model (**d**), or to derive the equations describing the perturbation expansion using computer symbolic algebra (**e**). The differential equations for the mean-field $q$, correction $p$ and spatial covariance $g$ in the perturbation expansion comprise a sum of terms contributed by each of the model components. These expressions can then be evaluated numerically (**f**), or analysed further by human or machine (**g**). The predictions from simulations and mathematical methods can then be compared (**h**) and used in many ways to gain insights into the focal problem (see Figs. 3, 4 and 5)

description, which can easily be translated to the syntax understood by the computer code. For simplicity, our software assumes a translational invariant initial condition, because there are many ways in which the underlying geometry could be spatially heterogeneous (see the Methods section for more discussion of this point). Below we provide a step-by-step description of the application of the framework to derive the underlying equations for a dynamic landscape patch occupancy metapopulation model. We then use three case studies drawn from different areas of ecology and evolution to illustrate how the framework of Fig. 2 allows researchers to obtain deep analytical insights into important problems.

**Using the framework**. The use of our framework follows the workflow illustrated in Fig. 2. We demonstrate this with an example for which the perturbation expansion has previously been derived via other methods: a dynamic landscape patch occupancy metapopulation[15]. For simplicity, we will here consider the case without landscape correlations (the limit $\nu \to 0$, $\alpha\nu = $ constant in ref. [15]).

We begin by deriving the equations for the sub-model that describes the dynamics of the habitat patches themselves (without

considering whether they are occupied), which is explicitly solvable. We assume that individual patches are created, at random, at rate per unit area $r$, and destroyed (at random) at rate $\mu$. Within our framework, this can be represented by a spatiotemporal point process with a single-entity type (habitat patches, which we denote as species type '1'). The formal mathematical representation of this model is the generator

$$L = L_1^{\mathrm{IM}}(r) + L_1^{\mathrm{D}}(\mu),$$

where the generators $L^{\mathrm{IM}}$ and $L^{\mathrm{D}}$ have technical definitions in the Supplementary Note 1, section 1.3, and can be understood intuitively as representing an immigration process (patch creation) with intensity $r$, and a density-independent death process (patch destruction) with rate $\mu$. The subscript '1' states the species to which the process applies (in this case, habitat patches are the only 'species'). These are two examples of the reactant–catalyst–product processes that, for convenience, we have defined in Supplementary Note 1 section 1.3, but note that we have derivations for the general reactant–catalyst–product process (Supplementary Note 1 section 1.4)—we could specify the model explicitly in terms of such general processes, but this would be somewhat more long-winded than using the above shorthand.

We will now show the steps for deriving the equations for this model using our Model Constructor (Supplementary Note 2 section 2.3). First, load the Mathematica packages containing the pre-defined list of processes and for deriving the equations:

```
get["SSPPlibraryOfProcesses'"]
get["SSPPanalyticalExpressions'"]
```

and define the Mathematica vector describing the processes in the model:

```
processes={Immigration[1,r,1],DensityIndepen
dentDeath[1,mu,1]}
```

The two functions Immigration[] and DensityIndependentDeath[] are pre-defined representations of the corresponding processes (see Supplementary Table 3 for the full list of pre-defined processes, and Supplementary Note 2 sections 2.1.3 and 2.3.4 for details on how to define specific reactant–catalyst–product processes). The first argument '1' is the species label, the second argument ($r$ or $mu$) is the corresponding rate and the third process is a (redundant) overall prefactor for the rates, included for consistency with other processes.

The quantity $H_q$ (the right-hand side for the differential equation for the mean-field patch density) is given by the Mathematica command:

```
HQfALL[{q,p,g},processes,1]
>r - mu q[1]
```

where {q,p,g} specifies the names of the variables Mathematica will use to denote the mean-field density, stochastic correction and correlation functions, processes is the vector defining the model given above, and '1' is the species label. To obtain the quantities $H_p$, which requires an integration in Fourier space, and $H_g$, we need to define the spatial dimension, and specify the name of the integration variable and the name for the frequency which is the argument of $g$:

```
HPfALL[{q,p,g}, processes, 1, k, 2]
>0
HGfALL[{q,p,g}, processes, 1, 1, k]
>0
```

For HPfALL, the argument '1' refers to the species label (of which there is only one in this simple model), and the argument '2' refers to the spatial dimension (which needs to be specified because HPfALL will in general include an integral over space). In HGfALL, both arguments '1' denote species labels, of which there need to be two because this quantity relates to correlation functions. The argument 'k' in both functions specifies the name of the variable that denotes the frequency, which appears as an argument to the Fourier transforms. Thus, our framework has derived the following equations for the model:

$$\frac{dq}{dt} = r - \mu q$$

$$\frac{dp}{dt} = 0$$

$$\frac{dg}{dt} = 0,$$

which are exact because this model is a simple immigration-death model, with no nonlinear processes (and hence no correction to mean-field) and no processes that introduce spatiotemporal correlations.

Adding metapopulation dynamics to this dynamic landscape, we now have two entity types: unoccupied patches (type 1), that can be colonised by occupied patches (type 2) (Fig. 2a, b). The dynamics are now represented by the following generator

$$L = L_1^{IM}(r) + L_{21}^{I}(c) + L_{12}^{CT}(e) + L_1^{D}(\mu) + L_2^{D}(\mu),$$

where the last two terms represent patch of either type being destroyed (with the same rate $\mu$), and the first one represents the fact that patches are unoccupied (type 1) when they are created. The second and third terms describe the metapopulation dynamics. The second term represents colonisation, where occupied patches turning an unoccupied patch into an occupied patch with kernel $c$ (i.e., the colonisation rate is $c(r)$ if the two patches are separated by distance $r$), and has superscript 'I' (denoting 'Infection') because this is functionally equivalent to type 2 patches 'infecting' type 1 patches. The third term corresponds to spontaneous changes in type (superscript 'CT') from type 2 to type 1, representing extinction of occupied patches.

These processes are represented in Model Constructor by the Mathematica process variable

```
processes={Immigration[1,r,1],Infection[2,1,
c,$c,1],
ChangeInType[1,2,e,1],
DensityIndependentDeath[1,mu,1],DensityInde
pendentDeath[2,mu,1]}
```

where the argument to the Infection process represents the kernel $c$, and the argument $c represents its Fourier transform. The equations for the quantities $q, g, p$ can again be obtained using the HQfALL, HGfALL, HPfALL functions. For example, $dq_1/dt$, $dp_2/dt$ and $dg_{12}(k)/dt$ are, respectively, given by the outputs of the following commands:

```
HQfALL[{q,p,g},processes,1]
>r - mu q[1] + e q[2] - q[1] q[2] $c[0]
HPfALL[{q,p,g},processes,2,k,2]
>2π∫₀^∞ k g[1,2,k] $c[k] dk - e p[2] - mu p[2]
+ (p[2] q[1] + p[1] q[2]) $c[0]
HGfALL[{q,p,g},processes,1,2,k]
>-e g[1,2,k] - 2 mu g[1,2,k] + e g[2,2,k]
+ g[1,1,k] q[2] $c[0] - g[1,2,k] q[2] $c[0]
+ g[1,2,k] q[1] $c[k] - g[2,2,k] q[1] $c[k]
- q[1] q[2] $c[k]
```

These equations are identical to those derived in ref. [15] for the case of an uncorrelated landscape (see the Methods section for the full set of differential equations). Our Mathematica toolbox contains tools for readily computing expressions for these quantities at equilibrium, as well as evaluating them numerically (see Supplementary Note 2 section 2.3).

Simulation code is obtained by the 'Model simulator' toolbox (Supplementary Note 2 section 2.2). The model is specified using a very similar markup to that for 'Model Constructor', except that parameters are given numerical values rather than symbolic names, and kernels have to be assigned specific functional forms. For instance, when colonisation has a top hat kernel with strength $\tilde{c}(0) = 1.5$ and length scale 3, and the other parameters are $(r, \mu, e) = (1.1, 0.9, 0.5)$, the syntax for Model Simulator is specified in file modelCaseStudy1.txt containing the text:

```
Immigration[1.1, 1]
Infection[2, 1, tophat[1.5, 3]]
ChangeInType[1, 2, 0.5]
DensityIndependentDeath[1, 0.9]
DensityIndependentDeath[2, 0.9]
```

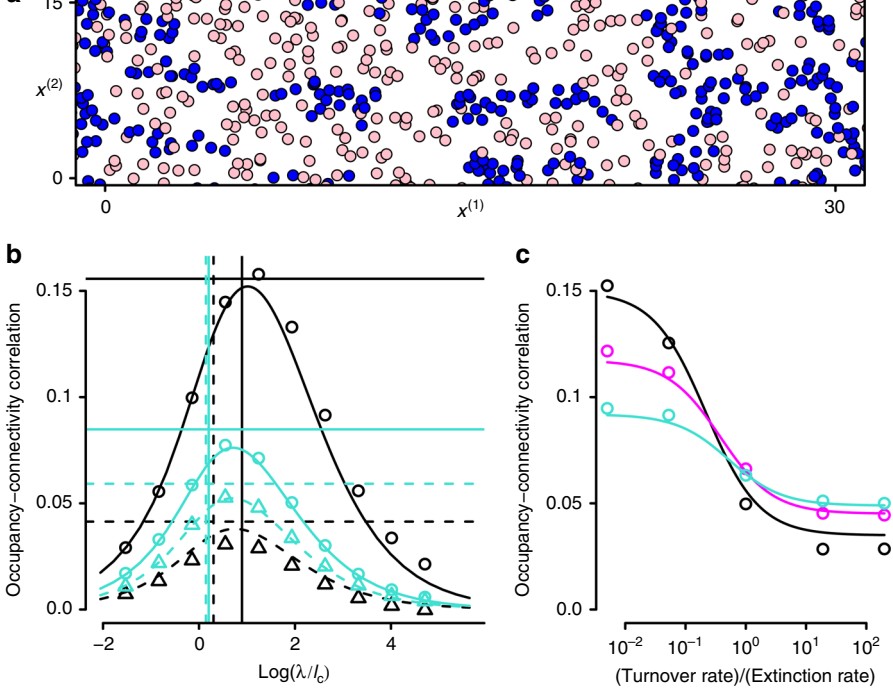

**Fig. 3** Occupancy–connectivity correlation in the dynamic landscape metapopulation. The model is defined in Fig. 2b. Snapshot **a** illustrates that patches are more likely to be occupied (blue symbols) rather than unoccupied (pink symbols) when they are close to other patches **b**. Connectivity-occupancy correlation depends strongly on the ratio of the length scale $\lambda$ of the connectivity kernel to the length scale $l_c$ of the colonisation kernel. Lines show analytical predictions and symbols are simulation results. Colonisation and connectivity kernels are chosen to have the same standard deviation when $\lambda = l_c$. The highest correlation occurs when $\lambda$ is 3–4 times larger that $l_c$. The optimal length scale depends only weakly on whether the landscape is static (solid lines, circles) or dynamic (dashed lines, triangles), or whether occupancy $P_0$ is high ($P_0 = 0.667$, cyan) or low ($P_0 = 0.167$, black). The correlation for the exponential connectivity kernel with the optimal choice of length scale is within 10–20% of the horizontal lines, which represent the occupancy–connectivity correlation for the optimal connectivity kernel shape $\tilde{h}_*$. The choice $\lambda = \lambda_*$ (vertical lines), where the connectivity kernel has the same standard deviation as the optimal connectivity kernel $h_*$, gives a higher occupancy–connectivity correlation than $\lambda = l_c$. **c** Occupancy–connectivity correlation decreases when rate of patch turnover $\mu$ increases relative to patch extinction rate $e$. Lines are analytical approximation, and symbols are the results of simulations. Parameters $r$ and $\mu$ are chosen so that patch density $=1$ and mean-field occupancy $P_0$ is kept constant as $\mu/e$ changes; colour denotes $P_0 = 0.167$ (black); $P_0 = 0.333$ (magenta); $P_0 = 0.667$ (cyan). Error bars not shown as standard errors are smaller than the plotting symbols. Parameter values and kernels are given in the Methods section

Simulations of the model are run by passing, as a command-line argument, the name of the file containing this model definition to the simulation programme ppsimulator we provide (Supplementary Note 2 section 2.2).

Other kernel shapes are implemented within ppsimulator, but for the sake of computational efficiency kernels that represent interactions between individuals are truncated so as to be zero beyond a threshold distance. For example if 'tophat[1.5,3]' were replaced by 'truncatedGaussian[1.5,3]' then the colonisation strength would still be $\tilde{c}(0) = 1.5$, but the kernel would be a truncated Gaussian (i.e., zero beyond a certain threshold distance, by default three standard deviations and Gaussian otherwise) with standard deviation 3. For more details, see Supplementary Note 2 section 2.2.

**Optimal landscape connectivity**. Our first case study concerns the question in conservation biology of how to identify the most important habitat to conserve. A commonly used metric to assess the value of habitat patch to a network, and aid the design of nature reserves, is 'connectivity'[16]. A critical open question is how well connectivity predicts patch occupancy—that is, whether connectivity is a good way to identify valuable habitat. Connectivity is expressed as a weighted sum of the proximity to other

habitat patches in the form[16] $S(x_i) = \sum_j h(x_i - x_j)$, where $h$ is a 'connectivity kernel', and $x_i$ and $x_j$ are positions of the $i$th patch and $j$th patch, respectively. Commonly, $h$ is taken to be an exponential function $h = \exp(-[\text{distance between patches}]/\lambda)$, where $\lambda$ is usually chosen to equal the species' average dispersal distance. While this is inspired from metapopulation ideas, there is no underlying theory for how well connectivity $S$ predicts habitat occupancy, or whether this kernel is the optimal choice[17]. Our framework allows us to solve this problem and find the best-performing connectivity measure.

We start with the dynamic landscape patch occupancy metapopulation model on a landscape where habitat patches are ephemeral (Figs. 2a, b, c; 3a)[15]. We showed above in 'Using the framework' how to use our 'model constructor' software to compute the quantities $q$, $p$ and $g$ for this model. We used this to derive an analytical expression for the correlation between patch occupancy and connectivity $S$ for a general connectivity kernel $h$ (see the Methods section), which can be expressed in terms of spatial moments and therefore in terms of our quantities $q$, $p$ and $g$. Considering first an exponential connectivity kernel, we find that the standard choice of the length scale $\lambda$ is not optimal as it gives a much lower correlation than when $\lambda$ is 3–4 times larger (Fig. 3b). We further derived an expression for the optimal connectivity kernel, i.e., the one that maximises the

connectivity–occupancy correlation:

$$\tilde{h}_\star(\omega) = \frac{\tilde{c}(\omega)}{\tilde{c}(0)(\tilde{c}(0)r + \mu^2) - \mu(e+\mu)\tilde{c}(\omega)}.$$

Here and throughout this paper, a tilde represents Fourier transform; $\tilde{h}_\star(\omega)$ is the Fourier transform of the optimal connectivity kernel, $\tilde{c}(\omega)$ is the Fourier transform of the colonisation kernel and $\omega$ is a frequency (the Fourier conjugate variable to space). The parameters $r$, $\mu$ and $e$ are defined in the caption of Fig. 2 and have the same meanings as in 'Using the framework' above (i.e., rates of patch creation, destruction and extinction respectively). It is clear from this expression that, in general, this kernel will have a different shape from either the exponential function or the colonisation kernel. However, we find that an exponential connectivity kernel with the optimal choice of $\lambda$ performs nearly as well (Fig. 3b). Thus, we show that an exponential connectivity kernel is a reasonable choice for landscape design, but we suggest that its ability to identify important habitat would be improved if its standard deviation were set to match that of the optimal kernel (Fig. 3).

This means that the connectivity kernel should not just depend on the dispersal properties as is usually assumed[16], but also on the metapopulation occupancy and landscape turnover as well (Fig. 3b). Moreover, we find that connectivity–occupancy correlation becomes weaker when the landscape becomes more dynamic (i.e., patches are created and destroyed more rapidly, Fig. 3c)[18]. Also, for static landscapes (left side of Fig. 3c), the correlation is stronger when the mean patch occupancy is lower, but this trend is reversed for dynamic landscapes (right side of Fig. 3c). Thus, connectivity is of less use to biological conservation when either occupancy or habitat turnover is high[19].

**Genetic similarity**. Our second case study is in the field of population genetics. Molecular ecologists study patterns of genetic similarity and differentiation between populations or across space to infer the underlying processes that cause and maintain that genetic structure. A long-standing problem is to find tractable models where local density is regulated so that population dynamics are stable[20]. Many ingenious solutions to this problem include imposing constant local density artificially[21–23]. However, since the aim of these models is to relate pattern to process, a much preferable solution would be to do this via local density-dependent population dynamics. Such models are regarded as too 'hopelessly complicated'[24,25] to study using standard methods: even in the simplest models, it is highly nontrivial to obtain analytical expressions for genetic similarity, and the incorporation of more realistic processes (e.g., such as selection) poses even greater difficulty[24,25]. Our framework overcomes these obstacles, allowing us to consider local density-dependent population regulation explicitly, and to derive analytical expressions for how genetic similarity varies in space. We illustrate this with an example including local competition and limited dispersal, selection and mutation between alleles in neutral and non-neutral loci, and also adaptation to heterogeneous environments, which is important for example for tropical forests[22].

The model involves two habitat types, and haploid individuals with four different genotypes (one selective and one neutral locus, two alleles at each). Habitat patches appear independently as a Poisson process with rate $\kappa$ and vanish with rate $\mu$. All individuals have density-independent mortality (rate $m$) and density-dependent mortality with the same interaction kernel $c$ between all genotypes. Birth is density-independent and offspring are distributed relative to their parents with kernel $d$, but the fecundity depends on habitat availability (via the patch kernel $r$)

and the match between the habitat type and the genotype of the individual via the function $\phi^\tau = f_0 \cdot (1 \pm \tau)$, where $f_0$ is a constant base fecundity and $\tau$ is the strength of selection. Therefore, each habitat patch a distance $x$ away from the individual contributes $r(x) \cdot f_0 \cdot (1 + \tau)$ to the fecundity of that individual if allele in selective locus matches the type of habitat, and $r(x) \cdot f_0 \cdot (1 - \tau)$ otherwise. Offspring inherits their parent's genotype (i.e., reproduction is clonal) unless there is a mutation, which happens independently at each allele at rate $\nu$ per birth and specified by the function $\psi^\nu$. All model components are shown in Fig. 4a, and a typical snapshot of the dynamics is illustrated in Fig. 4b.

We are interested in the similarity functions $F_n(x)$ and $F_s(x)$ at the neutral and selective loci, respectively, i.e., the probability that two individuals separated by distance $x$ have the same allele at the locus in question. These can be computed from the probability densities to find specific individuals at this spatial separation. Using the following function

$$F_i(x) = \frac{\text{probability density for individuals at } y \text{ and } x+y \text{ with allele } i}{\text{probability density for any two individuals at } y \text{ and } x+y},$$

where $i = 1, 2$ denote alleles in neutral locus, and $i = A, B$ denote alleles in selective locus, the similarity functions become $F_n(x) = F_1(x) + F_2(x)$, $F_s(x) = F_A(x) + F_B(x)$. The probability densities in the equation above can be expressed in terms of the two-point correlation functions (second spatial moments) $k_{i,j}^{(2)}(x_i, x_j)$ that in turn can be expressed in terms of the quantities $q$, $p$ and $g$ in the perturbation expansion[11], i.e., $k_{i,j}^{(2)}(x) = q_i q_j + \epsilon^d(g_{i,j}(x) + q_i p_j + p_i q_j) + o(\epsilon^d)$.

Using our framework, we obtained analytical expressions for how genetic similarity depends on the distance between individuals (Methods) which show good agreement with numerical simulations, Fig. 4c. Genetic similarity at the selective locus is always higher than at the neutral locus, and also extends further in space due to having different length scales $l_n$ and $l_s$ (Fig. 4c), where $l_s$, $l_n$ are, respectively, the length scales of genetic similarity at the selective and neutral loci. Our analytical expressions for genetic similarity (shown in the Methods section) enable us to understand what controls the differences in these length scales. The results become particularly simple under the plausible assumption that mutation is rare ($\nu \ll 1$, see the Methods section):

$$\frac{l_s}{l_n} = \frac{1 + \frac{\tau^2 R}{2\nu}\frac{M}{\mu}(1 - r^2)}{1 + \frac{\tau^2 R}{2\nu}\frac{M}{\mu}(1 - r)},$$

where $R = \frac{N_e}{2q_h}$ is the ratio of the total density of individuals $N_e$ to the total density of habitat patches, $M = 4\nu q_h f_0 \tilde{r}(0)\tilde{d}(0)$ is the per-capita creation rate of mutants in the population, $\mu$ is the turnover rate of habitat patches and $r = (1 + \mu/M)^{-1/2}$. We see that the ratio of these length scales is controlled by just two parameter combinations: $\mu/M$, which quantifies how ephemeral the landscape is relative to the appearance of mutants, and $\frac{\tau^2 R}{2\nu}$, which will be numerically large when $\nu \ll 1$ unless selection is very weak or habitat patches vastly outnumber the organism. Similarity at the selective locus will resemble that at the neutral locus if mutation is common or selection weak, but will extend up to twice as far when habitat turnover is slow relative to mutation (Fig. 4d).

**Optimal foraging**. Our third case study originates from the field of movement ecology, in which a long-standing challenge is to understand the causes and consequences of movement behaviour on foraging efficiency[26,27]. In environments with patchily

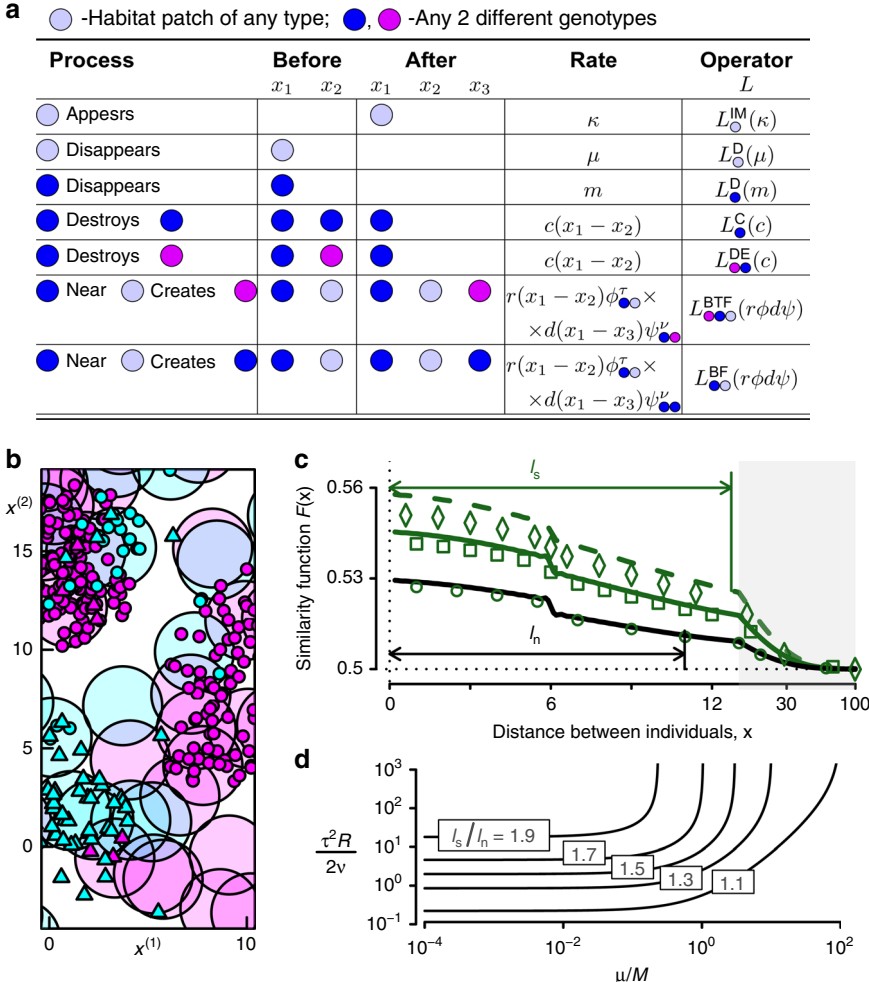

**Fig. 4** Spatial patterns of genetic differentiation in heterogeneous environments. **a** Graphical definition of the population model. **b** A snapshot of typical dynamics of the model: two types of habitat (light cyan or magenta discs), four genotypes of individuals (small symbols, colour denotes the preferred habitat of allele at selective locus, shape—triangle or circle—denotes allele at neutral locus), see the Methods section for values of parameters. **c** Similarity function (probability of allelic identity by state) at neutral (black) and selective (green) loci found theoretically (lines) and by simulations (symbols) have different length scales $l_n$ and $l_s$ (defined from the first moment of the similarity function). Parameters ($\tau =$ selectionstrength, $\mu =$ habitatturnover) equal (1, 0.5) for diamonds and the dashed line, (1, 1) for squares and solid green line, and (0.1, 1) for circles and solid black line. Here, at $x > 13$ a log-scale is used (grey area). **d** When mutation is rare ($\nu \ll 1$), the factor $l_s/l_n$ by which genetic similarity extends further at the selective locus than at the neutral locus depends only on the two parameter combinations $\frac{\tau^2 R}{2\nu}$ (with $R = \frac{N_e}{2q_h}$) and $\frac{M}{\mu}$. The length scales are most different when selection is strong, mutation rare, habitat static, and there are many individuals per habitat patch. Error bars not shown as standard errors are smaller than the plotting symbols. Parameter values are given in the Methods section

distributed resources, commonly observed behaviours involve slow foraging movements within patches, interrupted by fast exploratory movements between the patches[28]. Individual-based models are needed to account for stochasticity and resource heterogeneity in foraging[29], but there are several processes at work so it is very difficult from simulations to understand the key evolutionary drivers and ecological consequences of variation in foraging behaviour. Our framework solves this problem by providing simple analytical expressions that quantify the ecological factors determining the optimal foraging strategy.

We used the graphical model description (Fig. 5a) to construct a model in which aggregates of resources are continuously generated at new locations, while existing resource units decay, resulting from the consumer's point of view in an unpredictable resource distribution. The model consists of resources (targets), which are generated in clusters as follows. Target 'generators' appear as a Poisson process with intensity $b$, and disappear at rate $h$. During their lifetime, target generators create targets at rate $h\lambda$

at a distance from the generator determined by kernel $r$ (normalised so $\tilde{r}(0) = 1$). We assume $h \to \infty$ such that $h\lambda$ is constant, so that each generator instantaneously creates a Poisson distributed number of targets with mean $\sim\lambda$. Targets vanish at rate $\mu$. The forager moves as a jump process with jump kernel $c$ at rates $m_S$ and $m_F$, respectively, when in slow or fast mode. Foragers consume targets at rate $\gamma$ with kernel $f$. We assume the consumer has evolved to switch to a fast movement mode (at a rate we denote by $\alpha$) when it does not encounter new resources, and to switch back to slow movement mode after encountering a resource unit (Fig. 5a, b).

Our mathematical formalism yields a simple closed-form expression for the mean consumption rate (Fig. 5c), $\rho \approx \rho_0 + \epsilon^d \rho_1$, where $\rho_0 = \gamma\beta/\mu$ is the mean-field consumption rate, $\beta = \lambda b$, and the first-order correction to $\rho$ is

$$\rho_1 = \frac{\beta\gamma^2(\alpha\lambda\mu I_1 - \beta\gamma I_2)}{\mu(\beta\gamma + \alpha\mu)(\alpha + \mu)}, \tag{1}$$

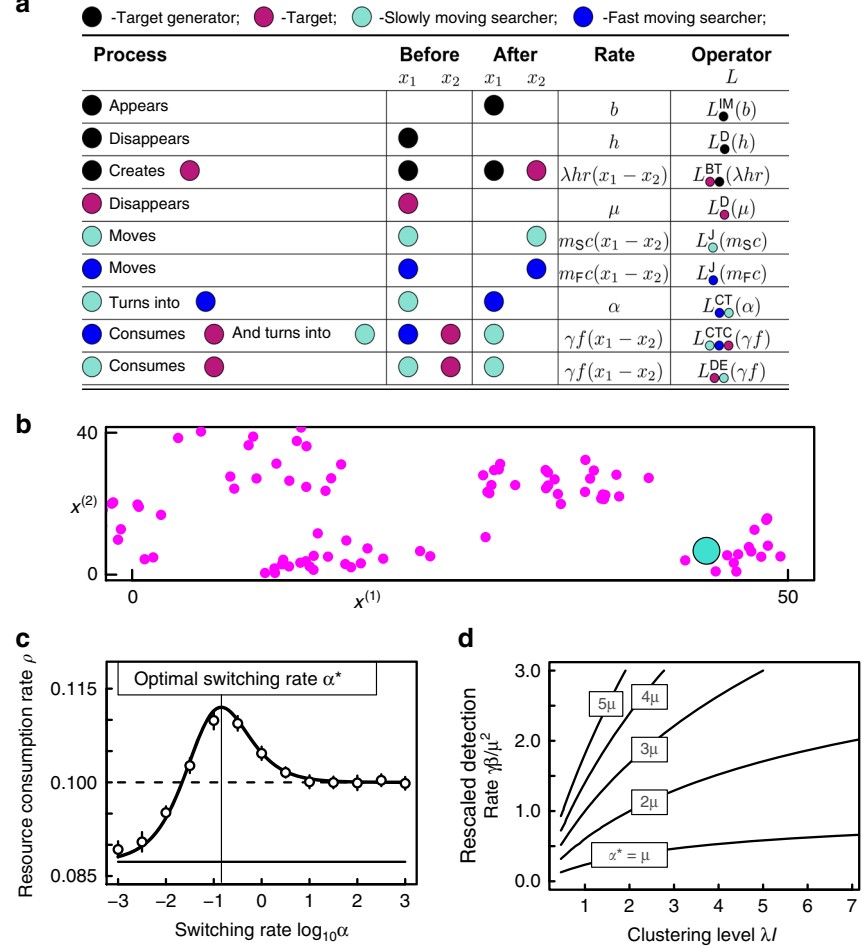

**Fig. 5 Optimal foraging model. a** shows a graphical definition of the model and **b** shows a simulation snapshot, illustrating a forager (large cyan circle) that has found a resource aggregate, and has switched to the slow movement mode to consume resources (small magenta circles). **c** shows how mean resource consumption rate $\rho$ varies from a totally stationary searcher ($\alpha = 0$; continuous horizontal line) through a searcher following the optimal rate $\alpha^*$ (vertical line) to a highly mobile searcher ($\alpha \rightarrow \infty$, dashed line). In **c**, the dots with error bars show simulation results, whereas the lines follow the analytical prediction of $\rho \approx \rho_0 + \epsilon^d \rho_1$ (see Eq. (1)). The contour lines in **d** show the optimal switching rate $\alpha^*$ as a function of the rescaled detection rate $\gamma\beta/\mu^2$ of the searcher and the clustering level $\lambda l$ of the targets. Error bars represent three standard errors. Parameters are shown in the Methods section

with $I_1 = \int \tilde{f}^2(\omega)\tilde{r}^2(\omega)d\omega$ and $I_2 = \int \tilde{f}^2(\omega)d\omega$. This expression behaves non-monotonically as a function of the switching rate, and is maximised at an intermediate value $\alpha^*$ (vertical line in Fig. 5c). This is because a consumer that remains continuously in the slow mode is not efficient in finding new resource aggregates, whereas a consumer that remains continuously in the fast mode misses the opportunity of elevated resource availability within aggregates.

We also find a simple expression for the optimal switching rate: $\alpha^* = \mu f\left(\frac{\rho_0}{\mu}, \frac{1}{\lambda I}\right)$, where $f(x,y) = xy + \sqrt{x(1+xy)(1+y)}$, $\lambda$ is the mean number of resource items per cluster, and $I = I_1/I_2$ ($0 < I < 1$) quantifies the spatial scale of the resource detection process relative to the resource cluster size. Thus, although we started with a model with nine processes, the optimal foraging strategy is only determined by the two parameter combinations $\frac{\rho_0}{\mu}$ and $\lambda I$ (Fig. 5d)—for example, an increase in resource acquisition rate has the same effect on the switching rate as the same proportionate increase in resource production rate. It therefore pays to spend a long time in the slow foraging mode (small $\alpha^*$) if the resources are highly aggregated (large $\lambda$) or if the consumer's resource detection efficiency ($\gamma$) is low, Fig. 5d. However, if fast movement requires more energy than the slow mode, we find that

$\alpha^* = 0$ when these energetic costs per unit time exceed a threshold (see the Methods section). This threshold value shows that it may be better to stay put and wait for resources to be generated locally, rather than to roam widely, when the resource consumption rate ($\rho_0$) is low, resource clusters contain few items ($\lambda$ small), or the spatial scale of clusters and/or resource detection is large.

## Discussion

Our framework is game-changing because (i) it makes mathematical approximations available for a very wide class of individual-based models, and (ii) this approximation is available to the non-specialist because it requires no laborious derivations on the part of the researcher. Since we have derived expressions for general reactant–catalyst–product model, and implemented these in Mathematica, a researcher interested in a particular model has only to convert their model into the markup understood by our software (see the Methods section and Fig. 2a–c) and the approximation is generated for them automatically. Without our method, mathematical results for a new model would only be available to a researcher with the skill and patience to spend weeks performing algebra without error. Our software also generates code for simulating the same model. The mathematical

validity of our approach is proved in Supplementary Note 1, and the validity of the approximation in these examples is shown by its close agreement with simulations in Figs. 3–5. Since our framework is based on a perturbation expansion, it is guaranteed to give a good approximation when the length scale of interaction kernels is large enough compared to the separation between individuals in the system[15]. One way to assess the accuracy of the approximation is to compare $p$ to $q$: if the first-order correction $p$ is comparable in magnitude to the mean-field density $q$, then approximation may not be very accurate for those parameter values because higher order corrections are likely to be important.

Our examples illustrate the conceptual and methodological advantages of analytical methods over simulations. In the first case study, we have a general expression for the best-performing connectivity kernel, valid for any colonisation kernel, whereas simulations can only explore specified examples and functional forms. In the third case study, we obtained directly an expression for the optimal foraging strategy, whereas in a simulation study the optimal strategy would have to be searched for separately for each parameter combination. We found in the second and third examples that the behaviour depends on a small number of parameter combinations, which effectively reduces the dimension of the parameter space of the problem. This means that the model behaviour can be explored more economically, and explained more intuitively, than in a simulation study in which each set of parameter values needs to be investigated independently. While simulations will always play an important role in studying spatial and stochastic models, our mathematical framework will facilitate much deeper understanding by making analytical results readily available for models that were previously thought to be too complex.

There are already platforms that allow simulation of spatial, stochastic individual-based models specified in a user-friendly way[10,30], but ours is the first framework that automatically performs mathematical analyses as well. We have formulated, analysed mathematically and written computer code for the general model containing any combination of reactant–catalyst–product processes. This general model can be used to describe a very large array of processes, so that a wide variety of models of interest in ecology and evolution are straightforward to construct and analyse using the unified framework presented here. Thus, our framework provides a long-sought analytical tool[8] with which many classical questions in ecology, evolution and other fields can be revisited to understand the role of space and stochasticity.

## Methods

**Derivation of the analytical framework**. Here, we give a summary of the approach and key results of our analytical framework; full details of the calculations are given in the Supplementary Notes.

We begin by classifying the participants in demographic processes into three types of individuals: (i) reactants (that are destroyed by the process); (ii) products (that are created by the process) and (iii) catalysts (that are unaffected by the process but whose presence affects the rate at which it occurs). A given demographic process is therefore characterised by integer numbers denoted by $P$, $R$ and $C$ that give number of agents in the groups 'reactants', 'products' and 'catalysts'. The group 'products' is characterised by the set $\mathbf{P}$ consisting of pairs of the type of agent $i$ and the location $x$ (from $d$−dimensional space $x \in \mathbb{R}^d$) assigned to the agent: $\mathbf{P} = \{\{i_1, x_{i_1}\}, \{i_2, x_{i_2}\}, \ldots, \{i_P, x_{i_P}\}\}$. Similarly, reactants are described by $\mathbf{R} = \{\{j_1, x_{j_1}\}, \{j_2, x_{j_2}\}, \ldots, \{j_R, x_{j_R}\}\}$, and catalysts are described by $\mathbf{C} = \{\{k_1, x_{k_1}\}, \{k_2, x_{k_2}\}, \ldots, \{k_C, x_{k_C}\}\}$. We introduce the function $r(\mathbf{P}, \mathbf{R}, \mathbf{C})$, which depends on types of agents and on distances between locations of agents used in $\mathbf{P}$, $\mathbf{R}$ and $\mathbf{C}$. The function $r$ determines interactions between agents by specifying those coordinates and indices which are the same in $\mathbf{P}$, $\mathbf{R}$, $\mathbf{C}$. For example, the 'Change in type' process where agents of type $j_1$ transform with rate $r_0$ into agents of type $i_1$ without changing their locations is described by: $\mathbf{P} = \{\{i_1, x_{i_1}\}\}$, $\mathbf{R} = \{\{j_1, x_{j_1}\}\}$, $\mathbf{C} = \varnothing$, and $r(\mathbf{P}, \mathbf{R}, \mathbf{C}) = r_0 \delta(x_{i_1} - x_{j_1})$.

For the purpose of this paper, it is convenient to work with cumulants $u$ instead of moments, as cumulants carry the same information as moments but lead to a more simple perturbation expansion[11]. We define the operator $Q^\Delta$ that determines

the full hierarchy of equations for cumulants:

$$\frac{\partial}{\partial t} u(\eta, t) = (Q^\Delta u)(\eta, t). \tag{2}$$

Here, all orders of cumulants are denoted by $u(\eta, t)$, where $\eta$ denotes any finite number of points in $d$−dimensional continuous space $\mathbb{R}^d$ occupied by agents of a specified type. For example, for a one-point configuration for an agent of type $m$ one has $\eta = \{x, m\}$, and the expression $u(\eta, t)$ gives a first-order cumulant $u_m^{(1)}(x, t)$ which is the density of agents $m$ at location $x$, denoted as $D_m(x, t) = u_m^{(1)}(x, t)$. For a two-point configuration $\eta = \{x, m; y, n\}$ one obtains a second cumulant $u(\eta, t) = u_{m,n}^{(2)}(x, y, t)$ that determines spatial covariance denoted $\mathrm{Cov}_{m,n}(x, y, t) = u_{m,n}^{(2)}(x, y, t)$.

Following ref. [11], we made the following steps in the derivation (for detailed derivations see Supplementary Note 1): (1) using notions of locally finite configurations, transform an agent-based model description into the dynamics of spatial moments; (2) using the correspondence between correlation functions and cumulants, obtain operator $Q^\Delta$ for the evolution of cumulants. We obtained:

$$
\begin{aligned}
(Q^\Delta u)(\eta, t) \;=\; \Bigg( & \int_{\mathbb{R}^{d(P+R+C)}} \mathrm{d}x_{i_1} \ldots \mathrm{d}x_{i_P} \mathrm{d}x_{j_1} \ldots \mathrm{d}x_{j_R} \mathrm{d}x_{k_1} \ldots \mathrm{d}x_{k_C}\, r(\mathbf{P}, \mathbf{R}, \mathbf{C}) \\
& \times \prod_{k=k_1}^{k_C} \left(1 + D_{x_k}^{\dagger(k)}\right) \left[\prod_{i=i_1}^{i_P}\left(1 + D_{x_i}^{\dagger(i)}\right) - \prod_{j=j_1}^{j_R}\left(1 + D_{x_j}^{\dagger(j)}\right)\right] V \Bigg)(\eta, t),
\end{aligned}
\tag{3}
$$

where we used the following operations $D_{x_m}^{(m)}$ and $D_{x_m}^{\dagger(m)}$, that can be considered as related to creation and annihilation operators:

$$\left(D_{x_m}^{(m)} u\right)(\eta) := u\left(\eta \cup x_m^{(m)}\right),$$

$$\left(D_{x_m}^{\dagger(m)} u\right)(\eta) := \sum_{y_m^{(m)} \in \eta} \delta\left(x_m^{(m)} - y_m^{(m)}\right) u\left(\eta \backslash x_m^{(m)}\right),$$

here $x_m^{(m)}$ denotes an agent of type $m$ at location $x_m$. In Eq. (3), $V$ can be thought of as a generating function:

$$V := (\exp^{*-1} u) * \left(D_{x_{j_1}}^{(j_1)} \ldots D_{x_{j_R}}^{(j_R)} D_{x_{k_1}}^{(k_1)} \ldots D_{x_{k_C}}^{(k_C)} \exp^* u\right),$$

where we used the operation $*$ that is defined below, also see ref. [11]. First, let $\Gamma_0$ denotes the set of all finite subsets $\eta$ of $\mathbb{R}^d$. For any functions $u, v$ on $\Gamma_0$, the $*$ operation is defined as:

$$(u * v)(\eta) := \sum_{\eta_1 \sqcup \eta_2 = \eta} u(\eta_1) v(\eta_2),$$

where the symbol $\bigsqcup$ denotes a disjoint union defined as

$$\sum_{\eta_1 \sqcup \eta_2 = \eta} u(\eta_1) v(\eta_2) = \sum_{\xi \subset \eta} u(\xi) v(\eta \backslash \xi),$$

thus, the symbol $*$ defines a convolution. Using these notations, the function $(\exp^* u)(\eta)$ is

$$(\exp^* u)(\eta) := \sum_{n=0}^{\infty} \frac{1}{n!} u^{*n}(\eta) = 1^*(\eta) + \sum_{n=1}^{\infty} \frac{1}{n!} \sum_{\eta_1 \sqcup \ldots \sqcup \eta_n = \eta} u(\eta_1) \ldots u(\eta_n),$$

where

$$1^*(\eta) = 0^{|\eta|} = 1_{|\eta|=0} = \begin{cases} 1, & \eta = \varnothing; \\ 0, & \eta \neq \varnothing. \end{cases}$$

The function $\exp^{*-1} u$ is the inverse with respect to the $*$ convolution, i.e., as was shown in e.g., [31], for any $u$ with $u(\varnothing) = 0$, there exist a function $\exp^{*-1} u$ such that $(\exp^{*-1} u) * (\exp^* u) = 1^*$,

$$\left(\exp^{*-1} u\right)(\eta) := \sum_{n=0}^{\infty} (-1)^n \left((\exp^* u) - 1^*\right)^{*n}(\eta).$$

The system of Eq. (2) cannot be solved exactly, because an equation for a cumulant of any order depends on higher order cumulants. In our derivations we follow the method from[11], which assumes that interaction between individuals is long ranged; details of derivations are presented in Supplementary Note 1, sections 1.1–1.3. Using this exact expression of the full hierarchy of moment equations for the general model, we apply the perturbation scheme[11,13] to derive a controlled approximation that gives asymptotically exact results when agents interact over large enough scales. It is assumed that spatial interactions between individuals depend on their separation $x$ (in $d$-dimensional space) according to interaction 'kernels' of the form $\epsilon^d f(\epsilon x)$ where $1/\epsilon$ is the typical length scale of interactions. In models with long-ranged interactions (i.e., where typical scale of interaction is large), the parameter $\epsilon$ is a small parameter that can be used to develop a perturbation expansion. It is this small parameter $\epsilon$ that makes this approximation controlled: the smaller $\epsilon$, the better leading terms in a perturbation expansion describe the exact solution.

We showed mathematically that for a model made up of a set general process defined by **P**, **R**, **C** and the interaction function $r_\kappa(\mathbf{P}, \mathbf{R}, \mathbf{C})$, the mean densities and spatial covariance (including autocovariance) satisfy the following expansion:

$$\text{density of species } m = q_m(\epsilon x, t) + \epsilon^d p_m(\epsilon x, t) + o(\epsilon^d),$$

$$\text{spatial covariance between species } m \text{ and n} = \epsilon^d g_{m,n}(\epsilon x_1, \epsilon x_2, t) + o(\epsilon^d),$$

where

$$\frac{dq_m(x, t)}{dt} = \sum_{\kappa \in \text{model components}} H^{(\kappa)}_{q_m}(x, t)$$

$$\frac{dp_m(x, t)}{dt} = \sum_{\kappa \in \text{model components}} H^{(\kappa)}_{p_m}(x, t)$$

$$\frac{dg_{m,n}(x_1, x_2, t)}{dt} = \sum_{\kappa \in \text{model components}} H^{(\kappa)}_{g_{m,n}}(x_1, x_2, t)$$

For a general (arbitrary) product–reactant–catalyst process $\kappa$, we derived the corresponding contributions $H^{(\kappa)}_{q_m}$, $H^{(\kappa)}_{p_m}$ and $H^{(\kappa)}_{g_{m,n}}$ to the differential equations for $q_m$, $p_m$ and $g_{m,n}$, see Eq. (2) in Fig. 1e. The expression $H^{(\kappa)}_{q_m}$ for the mean-field density $q_m(x, t)$ of an agent of type $m$ for an arbitrary spatially heterogeneous product–reactant–catalyst process is shown below:

$$H^{(\kappa)}_{q_m}(x, t) = \int_{\mathbb{R}^{d(P+R+C)}} dx_{i_1} \dots dx_{i_P} dx_{j_1} \dots dx_{j_R} dx_{k_1} \dots dx_{k_C} r_\kappa(\mathbf{P}, \mathbf{R}, \mathbf{C})$$
$$\times \left( \sum_{i=i_1}^{i_P} \delta_{mi}\delta(x_i - x) - \sum_{j=j_1}^{j_R} \delta_{mj}\delta(x_j - x) \right) \prod_{\beta_\zeta \in \{j_1, \dots, k_C\}} q_{\beta_\zeta}(x_{\beta_\zeta}). \quad (4)$$

Expression for $H^{(\kappa)}_{p_m}(x, t)$ and $H^{(\kappa)}_{g_{m,n}}(x_1, x_2, t)$ are shown below, while their derivations are presented in Supplementary Note 1.

$$H^{(\kappa)}_{p_m}(x, t) = \int_{\mathbb{R}^{d(P+R+C)}} dx_{i_1} \dots dx_{i_P} dx_{j_1} \dots dx_{j_R} dx_{k_1} \dots dx_{k_C} r_\kappa(\mathbf{P}, \mathbf{R}, \mathbf{C})$$
$$\times \left( \sum_{i=i_1}^{i_P} \delta_{mi}\delta(x_i - x) - \sum_{j=j_1}^{j_R} \delta_{mj}\delta(x_j - x) \right)$$
$$\times \left( \sum_{\beta_1 \in \{j_1, \dots, k_C\}} p_{\beta_1}(x_{\beta_1}, t) \prod_{\beta_\zeta \in \{j_1, \dots, k_C\}\backslash\beta_1} q_{\beta_\zeta}(x_{\beta_\zeta}, t) \right.$$
$$\left. + \sum_{\beta_1=j_1}^{k_C} \sum_{\beta_2=\beta_1+1}^{k_C} g_{\beta_1\beta_2}(x_{\beta_1}, x_{\beta_2}, t) \prod_{\beta_\zeta \in \{j_1, \dots, k_C\}\backslash\beta_1\backslash\beta_2} q_{\beta_\zeta}(x_\zeta, t) \right). \quad (5)$$

$$H^{(\kappa)}_{g_{m,n}}(x_1, x_2, t) = \int_{\mathbb{R}^{d(P+R+C)}} dx_{i_1} \dots dx_{i_P} dx_{j_1} \dots dx_{j_R} dx_{k_1} \dots dx_{k_C} r_\kappa(\mathbf{P}, \mathbf{R}, \mathbf{C})$$
$$\times \left\{ \left[ \sum_{i=i_1}^{i_P} \sum_{i'>i}^{i_P} (\delta_{mi}\delta_{ni'}\delta(x_i - x_1)\delta(x_{i'} - x_2) + \delta_{mi'}\delta_{ni}\delta(x_i - x_2)\delta(x_{i'} - x_1)) \right.\right.$$
$$- \sum_{j=j_1}^{j_R} \sum_{j'>j}^{j_R} (\delta_{mj}\delta_{nj'}\delta(x_j - x_1)\delta(x_{j'} - x_2) + \delta_{mj'}\delta_{nj}\delta(x_j - x_2)\delta(x_{j'} - x_1))$$
$$+ \sum_{k=k_1}^{k_C} \delta_{mk}\delta(x_k - x_1) \sum_{i=i_1}^{i_P} \delta_{ni}\delta(x_i - x_2) + \sum_{i=i_1}^{i_P} \delta_{mi}\delta(x_i - x_1) \sum_{k=k_1}^{k_C} \delta_{nk}\delta(x_k - x_2)$$
$$\left. - \sum_{k=k_1}^{k_C} \delta_{mk}\delta(x_k - x_1) \sum_{j=j_1}^{j_R} \delta_{nj}\delta(x_j - x_2) - \sum_{j=j_1}^{j_R} \delta_{mj}\delta(x_j - x_1) \sum_{k=k_1}^{k_C} \delta_{nk}\delta(x_k - x_2) \right]$$
$$\times q_{j_1}(x_{j_1}, t) \dots q_{j_R}(x_{j_R}, t) q_{k_1}(x_{k_1}, t) \dots q_{k_C}(x_{k_C}, t)$$
$$+ \sum_{i=i_1}^{i_P} \sum_{\beta_1 \in \{j_1, \dots, k_C\}} \left[ \delta_{mi}\delta(x_i - x_1) g_{\beta_1 n}(x_{\beta_1}, x_2) + \delta_{ni}\delta(x_i - x_2) g_{m\beta_1}(x_1, x_{\beta_1}) \right]$$
$$\times \prod_{\beta_\zeta \in \{j_1, \dots, k_C\}\backslash\beta_1} q_{\beta_\zeta}(x_{\beta_\zeta}, t)$$
$$- \sum_{j=j_1}^{j_R} \sum_{\beta_1 \in \{j_1, \dots, k_C\}} \left[ \delta_{mj}\delta(x_j - x_1) g_{\beta_1 n}(x_{\beta_1}, x_2) + \delta_{nj}\delta(x_j - x_2) g_{m\beta_1}(x_1, x_{\beta_1}) \right]$$
$$\times \prod_{\beta_\zeta \in \{j_1, \dots, k_C\}\backslash\beta_1} q_{\beta_\zeta}(x_{\beta_\zeta}, t) \right\}. \quad (6)$$

Using expressions $H^{(\kappa)}_{q_m}(x, t)$, $H^{(\kappa)}_{p_m}(x, t)$ and $H^{(\kappa)}_{g_{m,n}}(x_1, x_2, t)$, it is straightforward to show that values of quantities $q_m(\epsilon x, t)$, $\epsilon^d p_m(\epsilon x, t)$ and $\epsilon^d g_{m,n}(\epsilon x_1, \epsilon x_2, t)$ calculated using the given kernels $a$ are identical to the values of quantities $q_m(x, t)$, $p_m(x, t)$ and $g_{m,n}(x_1, x_2, t)$ calculated using the scaled kernels $a_\epsilon$, $a_\epsilon(x - y) = \epsilon^d a(\epsilon(x - y))$.

**Computer code**. The analytical software 'Model Constructor' is developed in Mathematica[32]. The code first requires a user to specify the model in terms of products, reactants and catalysts, and the interactions. Then, using Fourier transform and a simplifying assumption of translational invariance, the code uses the input data to obtain the analytical expressions for $H^{(\kappa)}_{q_m}(x, t)$, $H^{(\kappa)}_{p_m}(x, t)$ and $H^{(\kappa)}_{g_{m,n}}(x_1, x_2, t)$ for a given system in 1D, 2D or 3D infinite space. Also, the code can write down these analytical expressions in real space and in Fourier space into .tex file. The analytical software has been checked to reproduce the results for 15 specific processes (see Supplementary Notes) that were derived analytically.

The expressions produced by Model Constructor assume translational invariance, i.e., that the agents do not interact with a spatially heterogeneous

extrinsic environment, and the expressions are averaged over initial conditions that are translationally invariant. This is not due to a limitation of the underlying mathematical framework, as Eqs. (4), (5), (6) do not make this assumption. However, in the absence of translational invariance the equations for $g$ and $p$ involve two spatial co-ordinates rather than one, and as a result are much more challenging to solve both analytically and numerically. Any analytical solution would require exploiting whatever symmetries are present in the initial condition, which depends on the details of each individual case. While we are studying some specific cases without translational invariance (work in progress), software that allows spatial heterogeneity to be specified in a general way is beyond the scope of this paper.

The simulation software 'Model simulator' is a C-programme for simulating continuous-time point processes. Each point is associated with a coordinate and a discrete species attribute. Points are located either on 1D or 2D torus space. The user defines the set of processes and the initial configuration after which the simulator runs the Gillespie algorithm[33] in such a way that the information of point locations are taken into account, i.e., the system is not assumed to be well-mixed. The state of the configuration can be outputted at user-defined constant time intervals. Auxiliary R-functions are provided for calculating summary statistics and creating figures and animations based on the simulation. Input for the simulator is given by means of text files and few command line arguments. Output of the simulator is written in text files. The simulation software has been checked to replicate the results that we obtained earlier using more specific implementations (e.g. ref. [11]) that were fully independent of the current implementation (e.g., coded in different programming languages).

In all systems studied in this paper results of both types of software, analytical and simulation, match each other in the way that the mathematical theory suggests. The detailed derivations of underlying mathematical expressions, and the detailed tutorials for analytical and simulation software are shown in Supplementary Notes.

**Case studies**. Model Constructor was used to obtain the analytical results used in the case studies, and for representative parameter values these were compared with simulation results obtained using Model Simulator. The toolboxes and files containing the markup for the case studies are included at Figshare, https://doi.org/10.6084/m9.figshare.9633161. We give here the essential details for obtaining the case study results used in the paper; further details are given in Supplementary Notes 3, 4 and 5.

**Optimal landscape connectivity model**. For case study 1, the model is defined verbally in Fig. 2a and the processes are listed in Fig. 2b.

Perturbation equations: The section 'Using the framework' in the main paper contains a full description of the steps required to derive the following expressions for $q_i$ and $g_{ij}$ ($p_i$ is not needed for the rest of the calculation):

$$\frac{dq_1}{dt} = r - \mu q_1 + eq_2 - q_1 q_2 \tilde{c}(0)$$
$$\frac{dq_2}{dt} = -eq_2 - \mu q_2 - q_1 q_2 \tilde{c}(0)$$
$$\frac{d\tilde{g}_{11}}{dt} = -2\mu g_{11} + 2eg_{12} - 2g_{11}q_2\tilde{c}(0) - 2g_{12}q_1\tilde{c}(\omega)$$
$$\frac{d\tilde{g}_{12}}{dt} = (g_{11} - \tilde{g}_{12})q_2\tilde{c}(0) + (\tilde{g}_{22} - \tilde{g}_{12})(e - q_1\tilde{c}(0)) - 2\mu\tilde{g}_{12} - q_1 q_2 \tilde{c}(\omega)$$
$$\frac{d\tilde{g}_{22}}{dt} = -\frac{d\tilde{g}_{11}}{dt} - 2\frac{d\tilde{g}_{12}}{dt},$$

Setting $dq_1/dt = 0 = dq_2/dt = d\tilde{g}_{11}/dt = \dots$ etc., and solving (which can be performed by Mathematica), we find the following equilibrium densities and correlation functions that will be needed later:

$$q_1^* = (1 - P_0)Q_0$$
$$q_2^* = P_0 Q_0$$
$$g_{12} + g_{22} = \frac{Q_0 P_0(1 - P_0)}{\phi(\omega)(1 + \rho) - (1 - P_0)}$$

where

$$P_0 = 1 - \frac{\mu(e + \mu)}{\tilde{c}(0)r}$$
$$Q_0 = \frac{r}{\mu}$$
$$\phi(\omega) = \frac{\tilde{c}(0)}{\tilde{c}(\omega)}$$
$$\rho = \frac{1 - P_0}{1 + \frac{e}{\mu}}$$

Derivation of connectivity–occupancy correlation: We are interested in the product-moment correlation $R$ between occupancy $\sigma(x)$ ($= 1$ if there is an occupied patch at $x$ and 0 if there is not) and connectivity $S(x) = \sum_{y \in \gamma_A} h(x - y)$, where $\gamma_A$ is the set of all patches in the landscape whether occupied or not. This

can be defined as

$$R = \frac{\left\langle \left( \sigma(x) - \langle \sigma(x) \rangle_{x \in \gamma_A} \right) \left( S(x) - \langle S(x) \rangle_{x \in \gamma_A} \right) \right\rangle_{x \in \gamma_A}}{\left[ \left\langle \left( \sigma(x) - \langle \sigma(x) \rangle_{x \in \gamma_A} \right)^2 \right\rangle_{x \in \gamma_A} \left\langle \left( S(x) - \langle S(x) \rangle_{x \in \gamma_A} \right)^2 \right\rangle_{x \in \gamma_A} \right]^{1/2}},$$

where $\langle \cdot \rangle_{x \in \gamma_A}$ denotes an average over all patches in the landscape, i.e., $\langle Z(x) \rangle_{x \in \gamma_A} = \frac{\sum_{x \in \gamma_A} Z(x)}{|\gamma_A|}$. These averages were expressed as spatial moments (Supplementary Note 3 section 3.4), which were expanded using the perturbation expansion to leading order in $\epsilon^d$ were retained. Expressions for $q$ and $g$ above were then substituted into the expression for $R$ to give

$$R = \frac{\int \tilde{h}(\omega) \left( g_{12}(\omega) + g_{22}(\omega) \right) \mathrm{d}\omega}{\sqrt{Q_0^3 P_0 (1 - P_0) \int \tilde{h}^2(\omega) \mathrm{d}\omega}}. \tag{7}$$

$$= \frac{\int_{\mathbb{R}^d} \frac{\tilde{h}(\omega) Q_0 P_0 (1 - P_0)}{\phi(\omega)(1 + \rho) - (1 - P_0)} \mathrm{d}\omega}{\left( Q_0^3 P_0 (1 - P_0) \int_{\mathbb{R}^d} \tilde{h}^2(\omega) \mathrm{d}\omega \right)^{\frac{1}{2}}} \tag{8}$$

Proof that landscape turnover weakens connectivity–occcupancy correlation: From Eq. (7), we compute $\partial R / \partial \rho$, keeping the mean patch occupancy $P_0$ and mean patch density $Q_0$ constant:

$$\frac{\partial R}{\partial \rho} = -\frac{1}{\left( Q_0^3 P_0 (1 - P_0) \int_{\mathbb{R}^d} \tilde{h}^2(\omega) \mathrm{d}\omega \right)^{\frac{1}{2}}} \int_{\mathbb{R}^d} \tilde{h}(\omega) \frac{\phi(\omega) Q_0 P_0 (1 - P_0)}{(\phi(1 + \rho) - (1 - P_0))^2} \mathrm{d}\omega.$$

This expression is always negative.

Kernel that maximises correlation: We take a variational approach and write $\tilde{h}(\omega) = \tilde{h}_\star(\omega) + \nu \delta(|\omega| - |\omega_1|)$, where $\delta$ is a Dirac Delta function. If the correlation is maximal when $h = h_\star$, then $\partial R / \partial \nu = 0$ for all $\omega_1$, which leads to

$$\tilde{h}_\star(\omega) = \frac{1}{\phi(\omega)(1 + \rho) - (1 - P_0)}.$$

Variance and standard deviation of the optimal kernel: For any rotationally symmetric function $M(x)$ with variance $V_M = \int_{\mathbb{R}^d} x^2 M(x) \mathrm{d}x / \int_{\mathbb{R}^d} M(x) \mathrm{d}x$, the Taylor expansion of the Fourier transform of $M$ is

$$\tilde{M}(\omega) = \tilde{M}(0) \left( 1 - a V_M \omega^2 + o(\omega^2) \right),$$

where $a$ is a number that depends on spatial dimension, but not on $M$. Thus, $\tilde{c}(\omega) = \tilde{c}(0)(1 - a V_c \omega^2 + o(\omega^2))$, and expanding $\tilde{h}_\star$ gives

$$\tilde{h}_\star(\omega) = \frac{A}{\rho + P_0} \left\{ 1 - a V_c \omega^2 \frac{1 + \rho}{\rho + P_0} + o(\omega^2) \right\}.$$

Therefore, the variance of the optimal connectivity kernel $h_\star$ is

$$V_\star = V_c \left( \frac{1 + \rho}{\rho + P_0} \right)$$
$$= V_c \left( \frac{\tilde{c}(0)r + \mu^2}{\tilde{c}(0)r - e\mu} \right)$$

Thus, the standard deviation of the optimal kernel is $\lambda_\star = l_c \sqrt{(\tilde{c}(0)r + \mu^2)/(\tilde{c}(0)r - e\mu)}$, where $l_c$ is the standard deviation of the colonisation kernel.

Parameters used in Fig. 3: **a** $r = \mu = 0$, $c(x) = 4 \cdot$Tophat$(x, 1)$ where Tophat $(x, R)$ is a normalised tophat kernel with radius $R$, $e = 1$. **b** Connectivity kernel $h(x) = \exp(-x/\lambda)$, colonisation kernel $c(x) = 4 \cdot$Tophat$(x, l_c \sqrt{12})$ so that the colonisation and connectivity kernels have the same standard deviation when $\lambda = l_c$. Static landscape (solid lines, circles): $\mu = r = 0$, $e = 2$; dynamic landscape (dashed lines, triangles) $\mu = r = 2$, $e = 0$; **c** $r = \mu$ and $e = 2 - \mu$ so that patch density $= 1$ and mean-field occupancy $P_0 = 1 - \frac{e(\mu + e)}{r\tilde{c}(0)}$ is kept constant as $\mu/e$ changes; $c(x) = 4 \cdot$ Tophat$(x, 4)$; $h = \exp(-x/4)$.

**Genetic similarity model**. Full operator $L$ defining the model: Denoting two habitat types as species $s_1$ and $s_2$, and denoting species with four different genotypes as $s_3, s_4, s_5$ and $s_6$, the graphical definition of the model presented in Fig. 4a leads to the complete definition of the model by the following full operator $L$:

$$L = \sum_{i = s_1, s_2} \left( L_i^{\mathrm{IM}}(\kappa) + L_i^{\mathrm{D}}(\mu) \right) + \sum_{j = s_3, s_4, s_5, s_6} \left[ L_j^{\mathrm{D}}(m) + \sum_{k = s_3, s_4, s_5, s_6} L_{jk}^{\mathrm{DE}}(c) \right]$$
$$+ \sum_{i = s_3, s_4, s_5, s_6} \left[ \sum_{k_1 = s_3, s_4, s_5, s_6} \left( \sum_{k_2 = s_1, s_2} \phi^\tau_{k_1, k_2} \psi^\nu_{k_1, i} L_{i k_1 k_2}^{\mathrm{BTF}}(r, d) \right) \right],$$

where abbreviations have the following meaning: IM stands for Immigration, D for density-independent death, DE for death by external factor, BTF for birth to another type by facilitation (these processes are defined in Supplementary Note 1 section 1.3).

Probability densities: Using notations introduced in the main text, the probability density for any two individuals at locations $y$ and $x + y$ is considered as $\sum_{i, i'} k_{i, i'}^{(2)}(y, x + y)$, where the sum is taken over all possible pairs $(i, i')$ of genotypes (there are four possible genotypes: $\{A1, A2, B1, B2\}$). Probability density for individuals at $y$ and $x + y$ with allele $j$ in neutral locus (i.e., $j = 1, 2$) is defined as $\left( k_{Aj, Aj}^{(2)}(\bullet) + k_{Aj, Bj}^{(2)}(\bullet) + k_{Bj, Aj}^{(2)}(\bullet) + k_{Bj, Bj}^{(2)}(\bullet) \right)$, and with allele $\beta$ in selective locus (i.e., $\beta = A, B$) is defined as $\left( k_{\beta 1, \beta 1}^{(2)}(\bullet) + k_{\beta 1, \beta 2}^{(2)}(\bullet) + k_{\beta 2, \beta 1}^{(2)}(\bullet) + k_{\beta 2, \beta 2}^{(2)}(\bullet) \right)$, where in both cases $\bullet$ denotes locations of individuals $\bullet = \{y, x + y\}$.

Analytical expressions of similarity functions: We use Model Constructor to obtain expressions for $q$, $p$ and $g$ for the model, and hence expressions for the leading terms in the expansion for the two-point correlation functions (second spatial moments) $k_{i, j}^{(2)}$. To first-order in $\epsilon^d$, and from now on assuming we are in spatial dimension $d = 2$, the similarity functions $F_\mathrm{n}(x)$ and $F_\mathrm{s}(x)$ take the form

$$F_\mathrm{n}(x) = \frac{1}{2} + \frac{\pi}{4q^*} \int_0^\infty \frac{(1 - 2\nu)\tilde{d}(\omega)}{\tilde{d}(0) - (1 - 2\nu)\tilde{d}(\omega)} J_0(2\pi x \omega) \omega \mathrm{d}\omega,$$
$$F_\mathrm{s}(x) = F_\mathrm{n}(x) + \tau^2 \int_0^\infty \left( \frac{(1 - 2\nu)\tilde{d}(\omega)}{\tilde{d}(0) - (1 - 2\nu)\tilde{d}(\omega)} \right) \frac{\pi(1 - 2\nu) f_0 \mu \tilde{d}(\omega) \tilde{r}^2(\omega) / \tilde{r}(0)}{2 f_0 \kappa [\tilde{d}(0) - (1 - 2\nu)\tilde{d}(\omega)] \tilde{r}(0) + \mu^2}$$
$$\times J_0(2\pi x \omega) \omega \mathrm{d}\omega,$$

where $J_0(\omega)$ is Bessel function of the first kind of order zero and

$$q^* = \frac{2 f_0 \kappa \tilde{d}(0) \tilde{r}(0) - m\mu}{4\mu \tilde{c}(0)}$$

is the mean-field density for any single genotype.

In the limit of very rare mutations: $\nu \to 0$, the denominator in the integrals above is dominated by the behaviour for $\omega$ small. In small-$\omega$ limit in 2D one obtains $\tilde{d}(\omega) \approx \tilde{d}(0)(1 - 2\pi^2 \sigma^2 \omega^2) + o(\omega^3)$, where $\sigma^2$ is the variance of the kernel $d$. As a result, we have

$$F_\mathrm{n}(x) \approx \frac{1}{2} + \frac{1}{8\pi q^* \sigma^2} \int_0^\infty \frac{J_0(2\pi x \omega) \omega \mathrm{d}\omega}{R_\nu^2 + \omega^2}$$
$$= \frac{1}{2} + \frac{1}{8\pi q^* \sigma^2} K_0(2\pi x R_\nu)$$
$$= \frac{1}{2} + \frac{1}{2 N_e \pi \sigma^2} K_0(x / x_\mathrm{n}), \quad x_\mathrm{n} = \frac{\sigma}{2\sqrt{\nu}}$$

where

$$R_\nu^2 = \frac{\nu}{\pi^2 \sigma^2},$$

$N_e = 4q^*$, and we have assumed $x \ll \sigma$; $K_0$ is a modified Bessel function of the second kind of order zero. Similarly, when $\nu \to 0$, the similarity at the selective locus is

$$F_\mathrm{s}(x) \approx F_\mathrm{n}(x) + \frac{\tau^2}{8\nu q_\mathrm{h} \pi \sigma^2} \frac{K_0(x / x_\mathrm{n}) - K_0(x / x_\mathrm{s})}{\mu / M}, \quad x_\mathrm{s} = \frac{x_\mathrm{n}}{\sqrt{1 + \mu / M}},$$

where $q_\mathrm{h} = \kappa / \mu$ is the density of habitat patches of a single type, and $M = 4\nu q_\mathrm{h} f_0 \tilde{r}(0) \tilde{d}(0)$ is the rate with which an existing individual produces a new mutant.

Derivation of length scales: We define the length scale for a similarity function as the mean distance over which it decays to its asymptotic value at large distances $\left( \lim_{x \to \infty} F(x) = \frac{1}{2} \right)$:

$$l = \frac{\int_0^\infty x \left( F(x) - \frac{1}{2} \right) \mathrm{d}x}{\int_0^\infty \left( F(x) - \frac{1}{2} \right) \mathrm{d}x}.$$

According to this definition, using the small-$\nu$ expressions for $F_\mathrm{n}$ and $F_\mathrm{s}$ in terms of Bessel functions, we obtain the expression for the ratio of length scales presented in the main text.

Parameters used in Fig. 4: **b** $\nu = 0.01$, $\kappa = 0.05$, $\mu = 1$, $\tau = 1$, $f_0 = 1$, $m = 1$; $r(x) = 3.5 \times$ Tophat$(x, 2)$; $d(x) = 3 \times$ Tophat$(x, 2)$; $c(x) = 1.25 \times$ Tophat$(x, 3/20)$, where Tophat$(x, R)$ is a normalised tophat kernel with radius $R$. **c** $d(x) = r(x) =$ Tophat$(x, 6)$, and $c(x) =$ Tophat$(x, 20)$.

**Optimal foraging model**. The model is specified in section 'Optimal foraging' in the paper and Fig. 5a, and is defined by the operator

$$L = L_1^{\mathrm{IM}}(b) + L_1^{\mathrm{D}}(h) + L_{21}^{\mathrm{BT}}(h\lambda r) + L_2^{\mathrm{D}}(\mu)$$
$$+ L_3^{\mathrm{J}}(m_S c) + L_4^{\mathrm{J}}(m_F c) + L_{43}^{\mathrm{CT}}(\alpha) + L_{23}^{\mathrm{DE}}(\gamma f) + L_{342}^{\mathrm{CTC}}(\gamma f),$$

where the entity types are defined as follows: 1: resource target generators; 2: resource targets; 3: slow-moving consumer; 4: fast-moving consumer; the abbreviation BT stands for birth to another type, J for jump, CTC for change in type by consumption, other abbreviations were already explained above (the full definition is presented in Supplementary Note 1 section 1.3, see also Supplementary Tables 1–3). Parameter $h$ determines the rate at which target generators disappear; we will study the case where clusters of targets are created simultaneously by taking the

limit $h \to \infty$. Since each target generator creates resources at rate $h\lambda$, and has mean lifetime $1/h$, it generates a Poisson distributed number of resource targets with mean $\lambda$. This is a convenient method for generating clusters of individuals from an underlying process that generates individuals; a similar approach was used in ref. [15] to generate clusters of habitat patches.

Our interest is in the equilibrium rate $\rho^*$ by which each searcher consumes targets. This can be computed by solving the stationary state of the system under the condition that the density of searchers, denoted here by $A$, is constant. This yields $\rho^* = (\beta/k_2^* - \mu)k_2^*/A$, where $\beta = \lambda b$ is the appearance rate of targets, $k_2^*$ is the stationary density of targets and the expression in brackets is the rate at which a randomly selected target disappears due to consumption. As we are not interested in resource competition among multiple searchers, we consider the case of a single searcher, obtained technically by taking the limit $A \to 0$. Note that, in the main text, we have set $\epsilon = 1$, i.e., the length scale of the kernels $f$ and $r$ is the true biological one and has not been rescaled by $\epsilon$.

Resource consumption rate: Using Model constructor, the stationary density of targets in the mean-field approximation is $k_2^* \approx q_2^* = \beta/(A\gamma + \mu)$, and thus the consumption rate of targets by a single searcher is $\rho^* \approx \rho_0 = \gamma\beta/\mu$ independently of the parameter $\alpha$.

In the first-order approximation, the stationary density of targets is $k_2^* \approx q_2^* + \epsilon^d p_2^*$. The general expression for $p_2^*$ is given using 'The model constructor' toolbox: $p_2^* = \int_0^\infty \mathrm{IntF}(\omega)\mathrm{d}\omega$, where the integrand function $\mathrm{IntF}(\omega)$ is

$$\mathrm{IntF}(\omega) = -\frac{2\gamma\pi\omega\tilde{f}(\omega)}{(A\gamma + \mu)}\left(\tilde{g}_{23}(\omega) + \tilde{g}_{24}(\omega)\right).$$

Using 'The model constructor' toolbox to calculate the limit $h \to \infty$ (after which the results no longer depend on $h$) and to extract the leading term in the limit $A \to 0$, which is achieved by the following Mathematica commands:

```
IntF = IntF /. h -> (1/ih);
IntF = Normal[Series[IntF, {A, 0, 1}]];
IntF = Limit[IntF, ih -> 0]
```

We thus obtain:

$$\mathrm{IntF}(\omega) = \frac{2YAb\gamma^2\omega\pi\tilde{f}(\omega)^2}{\mu^2(b\gamma\lambda + \alpha\mu)Z},$$

where

$$
\begin{aligned}
Y &= b^2\gamma^2\lambda^3 + b\gamma\lambda^2\mu(2\alpha + m_\mathrm{F} + \mu) + \lambda\alpha\mu^2(\alpha + m_\mathrm{S} + \mu) \\
&\quad + (-m_\mathrm{F} + m_\mathrm{S})\tilde{r}(\omega)^2\mu^2\alpha(\alpha - \tilde{c}(\omega)l^2) \\
&\quad - \mu\tilde{c}(\omega)\lambda(b\gamma\lambda m_\mathrm{F} + \alpha m_\mathrm{S}\mu); \\
Z &= b\gamma\lambda(m_\mathrm{S} + \mu) + \mu(m_\mathrm{F} + \mu)(\alpha + m_\mathrm{S} + \mu) \\
&\quad - (b\gamma\lambda m_\mathrm{S} + \mu(\alpha m_\mathrm{F} + 2m_\mathrm{F}m_\mathrm{S} + m_\mathrm{F}\mu + m_\mathrm{S}\mu))\tilde{c}(\omega) \\
&\quad + m_\mathrm{F}m_\mathrm{S}\mu\tilde{c}(\omega)^2.
\end{aligned}
$$

The expressions for $\tilde{g}_{ij}$ are still rather unwieldy, but we can simplify them by considering the limit where in the fast movement mode the movements are very fast ($m_\mathrm{F} \to \infty$), and the slow movements are very slow ($m_\mathrm{S} = 0$). This is achieved by the Mathematica command

```
Simplify[Limit[IntF, mf -> Infinity]] /. ms -> 0
```

resulting in

$$\mathrm{IntF}(\omega) = \frac{2Ab\gamma^2\omega\pi\tilde{f}(\omega)^2(b\gamma\lambda^2 - \alpha^2\mu\tilde{r}(\omega)^2)}{\mu^2(\alpha + \mu)(b\gamma\lambda + \alpha\mu)}.$$

At this limit, the first-order approximation for the consumption rate of targets is $\rho^* \approx \rho_0 + \epsilon^d \rho_1$, where

$$\rho_1 = \frac{\beta\gamma^2(\alpha\lambda\mu I_1 - \beta\gamma I_2)}{\mu(\beta\gamma + \alpha\mu)(\alpha + \mu)},$$

and $I_1 = \int_0^\infty 2\pi\omega\tilde{f}(\omega)^2\tilde{r}(\omega)^2\mathrm{d}\omega$ and $I_2 = \int_0^\infty 2\pi\omega\tilde{f}(\omega)^2\mathrm{d}\omega$.

Optimal switching rate: The optimal switching rate is found by the value of $\alpha$ where $\partial\rho_1/\partial\alpha = 0$, which from the above expression for $\rho_1$ is

$$\alpha^* = \rho_0\frac{I_2}{\lambda I_1} + \sqrt{\mu\rho_0\left(1 + \frac{\rho_0}{\mu}\frac{I_2}{\lambda I_1}\right)\left(1 + \frac{I_2}{\lambda I_1}\right)}.$$

Threshold cost of fast foraging: During the fast-foraging mode, the resource acquisition rate is $\rho_0$, so it takes a time $1/\rho_0$ before the consumer finds a resource and switches to the slow mode. If the fast mode costs $\kappa$ resource units per unit time, then the rate of resource expenditure due to fast foraging is $\alpha\kappa/\rho_0$. The net resource gain is

$$\rho_0 + \epsilon^d\rho_1 - \alpha\kappa/\rho_0 = \rho_0 + \epsilon^d\frac{\rho_0\gamma(\alpha\lambda I_1 - \rho_0 I_2)}{(\rho_0 + \alpha)(\alpha + \mu)} - \frac{\alpha\kappa}{\rho_0},$$

where we used $\rho_0 = \gamma\beta/\mu$. This function has a maximum at $\alpha > 0$ (and,

correspondingly, its derivative with respect to $\alpha$ at $\alpha = 0$ is positive) unless

$$\kappa > \epsilon^d\frac{\rho_0\gamma}{\mu}\left[\lambda\int_{R^d}\left\{\tilde{f}(\omega)\tilde{r}(\omega)\right\}^2\mathrm{d}\omega + \left(1 + \frac{\rho_0}{\mu}\right)\int_{R^d}\tilde{f}^2(\omega)\mathrm{d}\omega\right],$$

in which case the fastest net resource acquisition is at $\alpha = 0$.

Parameters used in Fig. 5: Parameters not varied in the panels are set to $\gamma = 1$, $\beta = 0.01$, $\mu = 0.1$, $\lambda = 10$, $f(x) = r(x) = c(x) = \mathrm{Tophat}(x, 5)$. For panel c, the theoretical curve is calculated for $h = m_\mathrm{F} = 1/m_\mathrm{S} \to \infty$, whereas the simulations used $h = 1/m_\mathrm{F} = 1/m_\mathrm{S} = 1000$.

**Reporting summary**. Further information on research design is available in the Nature Research Reporting Summary linked to this article.

## Data availability
Simulation data used to generate Figs. 3, 4 and 5, created using the Model Simulator toolkit, are available from Figshare at https://doi.org/10.6084/m9.figshare.9632531.

## Code availability
Model Constructor and Model Simulator are released under the GNU Public License v. 2 and are available from Figshare at https://doi.org/10.6084/m9.figshare.9633161. Included with the software are sample files and code for running the case studies in this paper.

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

## Acknowledgements

O.O. and P.S. were supported by funding from the Academy of Finland (Grants nos. 1273253 and 250444 to O.O.) and the Research Council of Norway (CoE grant no. 223257). We would like to thank Mike Begon and Jane Rees for comments on the paper.

## Author contributions

S.C., O.O. and Y.S. designed the study; S.C., Y.S. and O.O. wrote the main text of the paper. Case study 1 was performed by S.C., case study 2 by Y.S. and S.C., case study 3 by O.O., S.C. and Y.S. In Supplementary Notes: Y.S. and D.F. wrote Supplementary Note 1 sections 1.1–1.3 (Introduction to mathematical framework and derivations for selected basic processes); Y.S. wrote Supplementary Note 1 sections 1.4 and 1.5 (derivations for a general process), Supplementary Note 2 sections 2.1 and 2.3 (Mathematica code and its tutorial); P.S. wrote Supplementary Note 2 section 2.2 (Simulation code and its tutorial); S.C. and Y.S. wrote Supplementary Note 3; Y.S. and S.C. wrote Supplementary Note 4; O.O. and Y.S. wrote Supplementary Note 5. All authors contributed to discussions and edited the paper.

## Competing interests

The authors declare no conflict of interest.
