## [Peer Review File · Nature Communications]

Reviewers' Comments:

Reviewer #1:

Remarks to the Author:

The authors present a new method for deriving analytical approximations of individual-based models, which then would make model analysis more comprehensive than by running the full individual-based simulation models.

I am not an expert in mathematical modelling, but I think I know the literature on analytical approximations quite well. The field got somewhat stuck in the „moment closure“ situation, and it seems the authors have made some major achievement regarding the state of the art in this field, for which I would like to commend the authors, I am really impressed.

Unfortunately, they are overselling their results way too much in terms of generality, referring to „individual-based models“ in general in the title, and to models „containing interactions of unlimited complexity“ in the abstract. In the article itself though, they are more precise by referring to „wide class of individual-based model“, but then this class is, in my understanding, actually a very narrow class, which represents probably much less than 1% of all individual-based models in the literature. Individuals are featureless points in space, and the only types of interaction is reproduction triggered by proximity in space, mortality caused by, e.g., predation, and moderation of these two processes by the presence of a third individual. Accordingly, the three example IBMs are so simple in structure that one would barely call them „individual-based“

The authors define IBMs via two features, discreteness and stochasticity, but they ignore two of the three elements of IBMs which are generally believed to define IBM: individuals are different (also during their life cycle), they interact locally, and they adapt their behaviour to their own state and that of their biotic and abiotic environment. Moreover, individuals have to cope with heterogeneous conditions in space.

I think this will make a great paper, if tuned down in the claims made, in a more specialized journal, like Theoretical Population Ecology, Mathematical Biosciences, or American Naturalist (or one of those Physics journals which publish „ecological“ papers), as the approach indeed is impressive with the considered type of simple models, which certainly play their role in science. But for the general readership of Nat. Comm., and also for individual-based or agent-based modellers in general, this work is too narrow in scope, I am sorry to say.

Reviewer #2:

Remarks to the Author:

This manuscript potentially represents a pivotal achievement in population ecology: a seamless and robust general engine for approximating spatially-explicit interactions within and among species in such a way that their dynamical properties can be formally evaluated. Such a platform would be enormously useful for the hardy band of ecologists grappling with the complications inherent in truly understanding the implications of spatial structure. Having said that, a key consideration is the modifier "potentially". It is a tall order to propose an analytical scheme that works universally for all applications and the gold standard for proof of this claim probably impossible to satisfy in a single manuscript. At minimum, one would want to demonstrate, as the authors have clearly done here, that the approximation procedure reliably reconstructs outcomes produced through highly replicated stochastic simulations of spatial point processes. Slightly more ambitiously, one would demonstrate that this is achievable in more than one case, as a minimal test of the assertion of generality. The

authors meet this criterion through 3 different test scenarios: (1) a consideration of the impact of connectivity on metapopulation persistence and the best way to score connectivity, (2) the effect of spatial processes on spatial variation in genetic structure, and (3) the impact of switching rate between dichotomous movement modes on rates of resource intake. These are all plausible examples of the kinds of spatial ecological processes current researchers struggle to represent and evaluate in moderately realistic spatial settings. Only time will tell, however, if the framework itself will hold up as a truly reliable and robust engine for any and all such systems. In the mean time, it seems a truly useful tool for advancing the case of spatial ecological processes, and that is in itself perhaps the most important attribute of good science.

Having said that, the core text of the manuscript suffers from the lack of a readable description of the step-by-step procedures by which the framework is applied. Key information is relegated to the figure legends, rather than being articulated and explained in less terse fashion in the text.

I would similarly suggest that none of the case studies is outlined in sufficient fashion in the online methods for even passionately enthusiastic readers to readily follow. As a result, one is forced to accept a great deal on faith without laboriously working through the massive Supplemental Materials.

I suggest that the authors consider working up one of the examples in much greater detail and relegating the other examples to the SI, as a means of both clarifying the steps in the framework as well as better clarifying the underlying logic of the formulations.

It might also be useful to compare the effectiveness of the approximation achieved using the proposed framework with the existing published results for identical point process analysed using moment-closure or full perturbation methodology. Perhaps the metapopulation model would be appropriate for this, since it has been exhaustively analysed.

In spite of these caveats, I congratulate the authors for an exciting and imaginative new approach!

John Fryxell
University of Guelph

Reviewer #3:

Remarks to the Author:

This paper presents a simulation platform for individual based models (IBMs), coupled with an analytical approximation of these models. The approach is applied to three case studies that span ecological and evolutionary questions.

There are two things I especially liked about the paper:

1. I agree entirely with the author's premise that analytical tractability limits the generality of spatial models. I am excited to see this paper because it starts to integrate simulation models with mathematical approaches to analyzing the equations that underpin these models.
2. The supporting documents are really impressive. There has been a lot of thought put into making this result a tool that naive users will (hopefully) be able to use.

There are also a few things that I fear will limit the impact of this work among biologists who might want to use the models:

1. The math behind these simple case studies is hard to follow. If the goal is to make the work accessible to ecologists with basic knowledge of differential equations (but no more), it would improve the paper dramatically to include an extremely simple case study or two - perhaps a contrasting pair in which one reduces to a mean field model and one does not - in addition to the more ecologically interesting ones.

2. Many (perhaps the majority of) ecological IBMs that I have seen are implemented specifically to look at dynamics in spatially heterogeneous environments, where spatial heterogeneity is fixed by factors outside the system (unlike the spatially variable resource patch model included in the paper). At first, I was disappointed because it almost seemed like those kinds of dynamics are outside the scope of this software. Then, it occurred to me that you could probably "hack" the program to get spatial heterogeneity, e.g., moving from x_1, y_1 to x_2, y_2 converts on product to another with different properties. I don't know if it would be worth building up a case study along these lines in the first cut of a paper, but it would certainly increase interest to ecologists who use IBMs to understand real-world (empirical) systems.

3. (minor) In some ways, the main message of the paper is lost in the complexity of the figures. I think the main message is that the analytical approximation looks a lot like the simulation output. This is shown in one panel of each figure, but you have to look carefully to see it among the diverse collection of panels describing the setup and outputs of the different models. I wonder if you want to have one figure that captures the essence of the paper, showing model output from IBMs and analytical approximations for each case study...?

In summary, I think that the paper will be exciting to applied mathematicians who work in theoretical ecology. The notion of merging computational and analytical approaches is also timely across a variety of scientific disciplines. I suspect the paper will be a bit inaccessible to quantitative ecologists who work with models but have only 1-2 years of university-level math training. Spending a a lot of time working through the supplemental manuals will help such an ecologist use the software, but the math would still be a black box without a simpler introductory example.

Reviewer #4:

Remarks to the Author:

See attached.

A unified framework for analysis of individual-based models in ecology and beyond

S. J. Cornell, Y. F. Suprunenko, D. Finkelstein, P. Somervuo and O. Ovaskainen

The aim of this manuscript is to provide a comprehensive mathematical and computational framework to analyse individual-based models of biological processes, with a key focus on problems in ecology. The authors tackle a very difficult problem, and their framework could provide a significant change in our ability to analyse individual-based models in biology. I have some major comments that I think should be addressed before the manuscript is suitable for publication.

1. I find the manuscript very difficult to read because information is put into so many different places. I would appreciate a more mathematical account of the modelling framework in the main text. Further, since the results of this manuscript are the theoretical pipeline, I'd rather details were placed in the results rather than in the methods. I think the user should have a good sense from reading the main text of what the theoretical pipeline does, where any assumptions come in and the extent to which they can be confident in the predictions of the analytically tractable model.
2. It is not clear what the novelty of the manuscript is. As far as I understand it, the approach to model coarse-graining was already presented by the authors, and presumably (I haven't checked) they provided some examples in their original publications of the use of their approach. Are the examples presented here different? What is new about the method presented in this manuscript? It is not necessarily a problem if the mathematical side is not new, because the computational implementation of their approach does seem novel and a potentially useful contribution to the research community. However, I think the authors should make clear the real contribution of this manuscript from the outset.
3. The authors should provide a clear discussion of the limitations of their approaches. For example, with the three case studies, does the good agreement between individual-based model and reduced model exist throughout parameter space? If not, how would a user be able to determine (except through repeated simulation) the magnitude of the error they should expect?
4. A significant contribution of the work is computational tools. However I did not see any details of how the software was developed. Is it fully tested and can the user be confident of the output? What are the (if any) implementation parameters? How do these affect the results of the model?
5. The simulation package seems only to deal with 1D models, or 2D models with doubly periodic boundary conditions. This is not very realistic. Could the authors extend their tool to deal with more appropriate domains and boundary conditions?
6. At several points in their derivation the authors mention translational invariance. Do all their models assume translational invariance? If so, could the approach be extended to include e.g. spatially heterogeneous initial conditions?
7. Does the use of the Gillespie algorithm for simulation preclude any deterministic behaviours in the individual-based model?
8. How efficient is the pipeline, compared to brute force simulation of the individual-based model?
9. Could the authors motivate their parameter choices in the case studies with measurements from the literature? Are the parameter values they have used biologically realistic?

Reviewer #1

The authors present a new method for deriving analytical approximations of individual-based models, which then would make model analysis more comprehensive than by running the full individual-based simulation models.

I am not an expert in mathematical modelling, but I think I know the literature on analytical approximations quite well. The field got somewhat stuck in the „moment closure“ situation, and it seems the authors have made some major achievement regarding the state of the art in this field, for which I would like to commend the authors, I am really impressed.

We thank very much for these encouraging comments.

Unfortunately, they are overselling their results way too much in terms of generality, referring to „individual-based models“ in general in the title, and to models „containing interactions of unlimited complexity“ in the abstract. In the article itself though, they are more precise by referring to „wide class of individual-based model“, but then this class is, in my understanding, actually a very narrow class, which represents probably much less than 1% of all individual-based models in the literature. Individuals are featureless points in space, and the only types of interaction is reproduction triggered by proximity in space, mortality caused by, e.g., predation, and moderation of these two processes by the presence of a third individual. Accordingly, the three example IBMs are so simple in structure that one would barely call them „individual-based“. The authors define IBMs via two features, discreteness and stochasticity, but they ignore two of the three elements of IBMs which are generally believed to define IBM: individuals are different (also during their life cycle), they interact locally, and they adapt their behaviour to their own state and that of their biotic and abiotic environment. Moreover, individuals have to cope with heterogeneous conditions in space.

In fact, our framework does not have the limitations the referee suggests, and we apologise for not explaining our framework clearly enough in the original manuscript. The wording in our abstract is “interactions of an unlimited **level** of complexity”, and while we maintain this to be accurate we realise we needed to be clearer about what we mean by this. In our framework, there is no limit to the number of reactants, catalysts, or products which can participate in an event (we are not limited to at most three participants, as the referee states). Also, our framework can contain any number of types of individual. The individuals are therefore not “featureless” because different entity types represent individuals with different features. Also, while an individual’s location is described by a spatial point, individuals do not behave in a “pointlike” way because they interact over a spatial scale via spatial kernels.

While we agree that our modelling approach does not cover every possible individual-based model, it does actually incorporate those two key elements of individual-based models that the reviewer mentions. First, as we consider a marked point process (i.e. we can consider an unlimited number of possible entity types), the individuals are characterized not only by their location but also any other characteristics, so they can be different, also during their life cycle. We actually utilize this feature of the modelling framework in our case studies: in case study 3 the consumer changes repeatedly between foraging rapidly or slowly during its lifetime; in case studies 2 and 3 individuals’ behavior depends on their genotype. Our current software (both simulation and analytical) can handle arbitrarily complex multivariate categorical marks. The mathematical framework can handle also continuously-valued marks, and incorporating such to our software presents one exciting challenge for the future.

Second, through the use of marks, the presented modelling framework allows the individuals to adapt their behavior to their own state and that of their biotic and abiotic environment. This is seen already in our examples, where in case study 2 fecundity depends on the individual's genotype and type of local habitat, whereas in case study 3 the foraging strategy depends on the proximity of food.

We have now included, in Box 1, a graphical depiction to help explain the generality of the class of models that can be described in our framework. We have also included a more detailed explanation (paragraph 1 of "the framework") to clarify what we mean by "unlimited level of complexity" and give an example to show that we can model complex processes with more than three participants.

Moreover, in common with other individual based model frameworks, the environment can be modeled as (non-biotic) agents. This provides a way to model a spatially heterogeneous environment, though in the original MS we quoted the simpler results that obtain when we average over stochastic realisations of the landscape so that the mean densities and correlation functions are translationally invariant. In response to comments by this and other referees, we now state how our framework can be applied when the system is not translationally invariant (see "Computer code" in Online Methods , and A.4.6 in the SI).

I think this will make a great paper, if tuned down in the claims made, in a more specialized journal, like Theoretical Population Ecology, Mathematical Biosciences, or American Naturalist (or one of those Physics journals which publish „ecological“ papers), as the approach indeed is impressive with the considered type of simple models, which certainly play their role in science. But for the general readership of Nat. Comm., and also for individual-based or agent-based modellers in general, this work is too narrow in scope, I am sorry to say.

We thank for this critical comment, which prompted us to revise our text to better explain both the generality and the limitations of our approach.

Reviewer #2

This manuscript potentially represents a pivotal achievement in population ecology: a seamless and robust general engine for approximating spatially-explicit interactions within and among species in such a way that their dynamical properties can be formally evaluated. Such a platform would be enormously useful for the hardy band of ecologists grappling with the complications inherent in truly understanding the implications of spatial structure. Having said that, a key consideration is the modifier "potentially". It is a tall order to propose an analytical scheme that works universally for all applications and the gold standard for proof of this claim probably impossible to satisfy in a single manuscript. At minimum, one would want to demonstrate, as the authors have clearly done here, that the approximation procedure reliably reconstructs outcomes produced through highly replicated stochastic simulations of spatial point processes. Slightly more ambitiously, one would demonstrate that this is achievable in more than one case, as a minimal test of the assertion of generality. The authors meet this criterion through 3 different test scenarios: (1) a consideration of the impact of connectivity on metapopulation persistence and the best way to score connectivity, (2) the effect of spatial processes on spatial variation in genetic structure, and (3) the impact of switching rate between dichotomous movement modes on rates of resource intake. These are all plausible examples of the kinds of spatial ecological processes current researchers struggle to represent and evaluate in moderately realistic spatial settings. Only time will tell, however, if the framework itself will hold up as a truly reliable and robust engine for any and all such systems. In

the mean time, it seems a truly useful tool for advancing the case of spatial ecological processes, and that is in itself perhaps the most important attribute of good science.

We thank the reviewer for these general encouraging comments. We fully agree with the viewpoint that the eventual utility of the framework presented here will be found out only with the help of applying it to a versatile set of case studies, which we indeed hope the researcher community to do in the future.

Having said that, the core text of the manuscript suffers from the lack of a readable description of the step-by-step procedures by which the framework is applied. Key information is relegated to the figure legends, rather than being articulated and explained in less terse fashion in the text. I would similarly suggest that none of the case studies is outlined in sufficient fashion in the online methods for even passionately enthusiastic readers to readily follow. As a result, one is forced to accept a great deal on faith without laboriously working through the massive Supplemental Materials. I suggest that the authors consider working up one of the examples in much greater detail and relegating the other examples to the SI, as a means of both clarifying the steps in the framework as well as better clarifying the underlying logic of the formulations.

We thank very much for this valuable comment, which inspired us to make a thorough re-structuring of the paper. As suggested by the reviewer, we now give a step-by-step account in Box 2 of the derivation of the underlying equations for one of the examples (the metapopulation case study), so that the reader does not need to go to the Supplemental Materials to see how to use our framework. We have re-structured the Results section, Online Methods, and figure captions so that the results can be more easily followed in the main manuscript, and that the calculations can be understood without reference to the Supplementary Information.

It might also be useful to compare the effectiveness of the approximation achieved using the proposed framework with the existing published results for identical point process analysed using moment-closure or full perturbation methodology. Perhaps the metapopulation model would be appropriate for this, since it has been exhaustively analysed.

A comparison of the type suggested by the referee has already been published for the Spatial Logistic Model, for which our new framework produces identical results to previous formalisms of the perturbation method (ref. 13). A comparison in a second model is not guaranteed to give a better indication of the performance of different approaches in any third scenario of interest to a reader. We therefore decided not to include such comparisons, and instead cite in the main text (“The Framework” paragraph 2) the published comparison between moment closures and the perturbation approach.

In spite of these caveats, I congratulate the authors for an exciting and imaginative new approach!

*John Fryxell
University of Guelph*

Reviewer #3

This paper presents a simulation platform for individual based models (IBMs), coupled with an analytical approximation of these models. The approach is applied to three case studies that span ecological and evolutionary questions. There are two things I especially liked about the paper:

1. I agree entirely with the author's premise that analytical tractability limits the generality of spatial models. I am excited to see this paper because it starts to integrate simulation models with mathematical approaches to analyzing the equations that underpin these models.

2. The supporting documents are really impressive. There has been a lot of thought put into making this result a tool that naive users will (hopefully) be able to use.

We thank the reviewer very much for these encouraging comments.

There are also a few things that I fear will limit the impact of this work among biologists who might want to use the models:

1. The math behind these simple case studies is hard to follow. If the goal is to make the work accessible to ecologists with basic knowledge of differential equations (but no more), it would improve the paper dramatically to include an extremely simple case study or two - perhaps a contrasting pair in which one reduces to a mean field model and one does not - in addition to the more ecologically interesting ones.

This important comment echoes a suggestion by Reviewer 2. As explained above in our response to Reviewer 2, we have worked out one of the examples (the metapopulation case study) in much more detail in the main text, showing how to derive the underlying differential equations with our framework. This example includes a sub-model where the mean-field is exact (an immigration-death model for the habitat patches) as well as the more complex model that includes colonization-extinction dynamics.

2. Many (perhaps the majority of) ecological IBMs that I have seen are implemented specifically to look at dynamics in spatially heterogeneous environments, where spatial heterogeneity is fixed by factors outside the system (unlike the spatially variable resource patch model included in the paper). At first, I was disappointed because it almost seemed like those kinds of dynamics are outside the scope of this software. Then, it occurred to me that you could probably "hack" the program to get spatial heterogeneity, e.g., moving from x_1, y_1 to x_2, y_2 converts on product to another with different properties. I don't know if it would be worth building up a case study along these lines in the first cut of a paper, but it would certainly increase interest to ecologists who use IBMs to understand real-world (empirical) systems.

Let us first note that extending the simulation software to include pre-defined spatial (or spatio-temporal) heterogeneity is straightforward, and we have applied such extensions in our ongoing research. For example, we extended the simulation software to study how different patterns of habitat fragmentation influence the dynamics and persistence of metacommunities (Rybicki et al., unpublished manuscript). However, as such extensions can be done in many different ways, incorporating a general module for external heterogeneity is beyond the scope of the present paper. Concerning the mathematical methods, our analytical results are valid in the spatially heterogeneous system – these results are shown explicitly in the main text of the revised manuscript (Box 1) and the Online Methods section. These equations can be studied analytically or numerically.

Concerning the analytical software, the approach presented in the original submission was indeed restricted to the case of a spatial homogeneity of initial conditions, translational invariance, and radially symmetric interaction kernels. This was not because the mathematical framework would be restricted to such cases, but to keep the presentation of the methods and results simple. For example, with the simplifying assumptions made, the two-point correlation function only depends

on distance between the two points (one number), not the coordinates of those two points (four numbers for two-dimensional space). However, we agree with the reviewer that for many interesting problems (such as invasion from small spatially restricted initial population, or externally defined environmental heterogeneity) the assumptions are restrictive, and we have now extended Online Methods and Supplementary Information to describe how the framework can be applied in the more general case, and mentioned this in the main text (The Framework, para 2; “Computer Code” in Online Methods; Section A.4.6 in SI).

3. (minor) In some ways, the main message of the paper is lost in the complexity of the figures. I think the main message is that the analytical approximation looks a lot like the simulation output. This is shown in one panel of each figure, but you have to look carefully to see it among the diverse collection of panels describing the setup and outputs of the different models. I wonder if you want to have one figure that captures the essence of the paper, showing model output from IBMs and analytical approximations for each case study...?

While the figures do serve to confirm that the simulations support the results of the analytical approach, our main purpose is to give examples to show that the analytical method can indeed be used to address many kinds of mathematically non-trivial and ecologically interesting problems. We think, therefore, that figures that illustrate the biologically interesting message should be the focus of the paper. Our approximation is based on an existing scheme which is known to be exact in a particular limit. While we do show one raw comparison between analysis and simulation (Fig. 1h), we think a figure that highlights this alone would be disingenuous because we can always choose parameters where the approximation and simulation results are in excellent agreement.

In summary, I think that the paper will be exciting to applied mathematicians who work in theoretical ecology. The notion of merging computational and analytical approaches is also timely across a variety of scientific disciplines. I suspect the paper will be a bit inaccessible to quantitative ecologists who work with models but have only 1-2 years of university-level math training. Spending a lot of time working through the supplemental manuals will help such an ecologist use the software, but the math would still be a black box without a simpler introductory example.

We thank for this summary and share the perspective of the reviewer. We hope that the more fully worked out example that we have added to the main paper will help making the methods accessible for a broad set of researchers.

Reviewer #4

The aim of this manuscript is to provide a comprehensive mathematical and computational framework to analyse individual-based models of biological processes, with a key focus on problems in ecology. The authors tackle a very difficult problem, and their framework could provide a significant change in our ability to analyse individual-based models in biology.

We thank for these encouraging general comments.

I have some major comments that I think should be addressed before the manuscript is suitable for publication.

1. I find the manuscript very difficult to read because information is put into so many different places. I would appreciate a more mathematical account of the modelling framework in the main text. Further, since the results of this manuscript are the theoretical pipeline, I'd rather details were

placed in the results rather than in the methods. I think the user should have a good sense from reading the main text of what the theoretical pipeline does, where any assumptions come in and the extent to which they can be confident in the predictions of the analytically tractable model.

We have restructured the manuscript to address this comment (which also echoes the comments by Reviewers 2 and 3). The theoretical pipeline is illustrated in Box 2, the assumptions and results for the case studies are given in the main text rather than figure captions, the key equations we derived for the general model are given in the Methods rather than the SI, and we give further discussion of the validity of the approximation (see later).

2. It is not clear what the novelty of the manuscript is. As far as I understand it, the approach to model coarse-graining was already presented by the authors, and presumably (I haven't checked) they provided some examples in their original publications of the use of their approach. Are the examples presented here different? What is new about the method presented in this manuscript? It is not necessarily a problem if the mathematical side is not new, because the computational implementation of their approach does seem novel and a potentially useful contribution to the research community. However, I think the authors should make clear the real contribution of this manuscript from the outset.

While we have applied the perturbation approach to some specific models in the past, the key novelty here is that we derive results for a general class of models. While in Ovaskainen et al. (2014) the mathematics needed to be worked out separately for each model component (say, density dependent mortality), in the current manuscript we present one meta-model (the general product-reactant-catalyst) model that generates a very general family of such model components as special cases (e.g. all of those used in our case studies). The new mathematical result in the present manuscript is the derivation of the mathematically exact perturbation expansion for the general meta-model. This new result has enabled us to build analytical software that works at the very general level: it allows the user not only to consider problems with pre-defined model components, but also to define novel model components by translating the novel model component into product-reactant-catalyst notation, after which analytical results for the newly defined model components will be immediately available.

We have revised the text to make the novelty of the present paper more clear, including a new Box 1 to explain the limitations of previous approaches and the assumptions, novelty and generality of our new solution.

3. The authors should provide a clear discussion of the limitations of their approaches. For example, with the three case studies, does the good agreement between individual-based model and reduced model exist throughout parameter space? If not, how would a user be able to determine (except through repeated simulation) the magnitude of the error they should expect?

This is a valuable point: the mathematical results quantify the asymptotic rate at which the results from the full individual-based model converge to the results of the analytical approximation, but they do not quantify the absolute error (the constant in front of the asymptotic rate). The validity of the perturbation approach has been explored and discussed in previous publications. The only definitive quantification of the accuracy of the approximation for a particular set of parameter values is to compare with simulations. However, a comparison of the mean-field result q to the first-order correction p does provide an indication of the degree of accuracy of the approximation, as higher order terms are likely to be unimportant if p is small relative to q . This is now discussed at the end of Discussion, para 1.

4. A significant contribution of the work is computational tools. However I did not see any details of how the software was developed. Is it fully tested and can the user be confident of the output? What are the (if any) implementation parameters? How do these affect the results of the model?

We now discuss this in a new section in the Online Methods (“Computer code”). The analytical software has been checked to reproduce the results that we have derived earlier using pen and paper (e.g. Ovaskainen et al. 2014), and the simulation software has been checked to replicate the results that we obtained earlier using more specific implementations (e.g. Ovaskainen et al. 2014) that were fully independent of the current implementation (e.g. coded in different programming languages). While it is never possible to fully exclude the possibility of mistakes in computer code, in our view the best “proof” of the validity of both types of software is that their results match each other in the way that the mathematical theory suggests (for comparisons between the analytical and simulation results that assess the rate of convergence, see Ovaskainen et al. 2014).

5. The simulation package seems only to deal with 1D models, or 2D models with doubly periodic boundary conditions. This is not very realistic. Could the authors extend their tool to deal with more appropriate domains and boundary conditions?

The analytical results are valid for infinitely large landscapes. To mimic such a case with the simulations, we have (in addition to running replicate simulations for increasingly large domain) implemented periodic boundary conditions. While it would be challenging to incorporate finite domains and different types of boundary conditions to the analytical methods, incorporating such to the simulation software is in principle straightforward. However, we have decided not to do so as we consider the analytical software and simulation software as a parallel set of tools and thus we wanted to keep them as comparable as possible.

6. At several points in their derivation the authors mention translational invariance. Do all their models assume translational invariance? If so, could the approach be extended to include e.g. spatially heterogeneous initial conditions?

This comment was similar to the one made by Reviewer 3. Concerning the analytical software, the approach presented in the original submission was indeed restricted to the case of a spatial homogeneity of initial conditions, translational invariance, and radially symmetric interaction kernels. This was not because the mathematical framework would be restricted to such cases, but to keep the presentation of the methods and results simple. For example, with the simplifying assumptions made, the two-point correlation function only depends on distance between the two points (one number), not the coordinates of those two points (four numbers for two-dimensional space). However, we agree with the reviewer that for many interesting problems (such as invasion from small spatially restricted initial population, or externally defined environmental heterogeneity) the assumptions are restrictive, and we have now extended the Supplementary Information (A.4.6) to describe how the framework can be applied in the more general case.”

7. Does the use of the Gillespie algorithm for simulation preclude any deterministic behaviours in the individual-based model?

If the reviewer means by “deterministic behaviors” processes such as “the individual moves at constant speed of 1 m/s to South-East during 8am-8pm of each day, but stays sedentary during the

night”, then implementing them through the Gillespie algorithm would either be impossible or very difficult – indeed, exact simulation of a system with both stochastic and deterministic processes in continuous time will in many cases be impossible. While, this sort of behavior could be implemented within our analytical framework by taking appropriate limits of stochastic processes, we would need to extend the software to non-rotationally-symmetric kernels and consider an elaboration of this sort to be beyond the scope of the current manuscript.

8. How efficient is the pipeline, compared to brute force simulation of the individual-based model?

The analytical pipeline corresponds to results that one would obtain for an infinitely large simulation domain and thus in this sense it is very efficient. The simulation pipeline is essentially a brute force simulation of the individual-based model. To make it computationally efficient, we have paid much attention for optimizing the code. This is the reason why we encourage the user to apply interaction kernels with finite support, as they allow the use of techniques that make the computational time scale as the first power (instead of the second power that would be achieved in a more straightforward implementation) of the number of individuals for simulating a fixed number of events, and it thus scales as the second power (instead of third power) of the number of individuals for simulating the model for a given duration of time.

9. Could the authors motivate their parameter choices in the case studies with measurements from the literature? Are the parameter values they have used biologically realistic?

The examples are meant to serve as qualitative illustrations, so they have not been tailored to any specific biological systems.

Reviewers' Comments:

Reviewer #3:

Remarks to the Author:

This manuscript goes a long way to addressing concerns about the earlier version of the same paper. I enjoyed reading it, and also found the methods much clearer. I especially appreciate the inclusion of a worked example (Box 2), and the extent to which the revised manuscript tones down claims of being a "global" solution to all individual based models.

I am sure the work will be a valuable contribution to our understanding of spatial ecology and population dynamics.

Now that I understand the manuscript better, I see that it solves one problem of individual-based simulation models, specifically (as the authors note) by providing numerical solutions rather than averaging over a number of replicate simulations, but does not solve another of their problems. The latter problem is the "black-box" nature of simulation models, which often have arbitrarily-specified parameters and hidden assumptions. Since we will rarely know the shape of many of the functions that go into these models, or the strength of the interactions among organisms, it would be easy to get results from this kind of model (or any simulation model) that depend mostly on the user's opinion, or desired output. I find that I prefer simpler models (simpler mathematically) because the assumptions are more transparent, and potentially more in line with the nature of ecological data. [This is more of a comment than a criticism.]

This said, I am impressed by the work that has gone into this paper, and I am interested to see how the work gets used after it is published.

A few minor comments:

1st equation in Box 1: I really want the 2nd term on the RHS to be negative, even though it is technically correct to have it be positive

Box 2, 2nd page: I do not understand the phrase "the name of the spatial frequency" - is this the name of a standard probability distribution or kernel used in the model?

I believe there is a typo in the legend for figure 2, where it says "length scale lambda of the connectivity kernel to the length scale l_c of the connectivity kernel" [should one "connectivity" be "dispersal"?)

Reviewer #5:

Remarks to the Author:

The authors introduce a new framework for the analysis of various individual-based models. The approach is then applied to three case studies from movement ecology, evolutionary ecology and conservation biology.

The approach is very interesting and has the potential to change the field. Therefore the manuscript deserves publication.

However, before that, there are still a few aspects that still need to be addressed:

1) The Framework is presented briefly, and then applied to three different case studies, which are not

always easy to understand (when reading the main manuscript).

a. For example, in the Optimal Landscape Connectivity case, the function $h^*(\omega)$ that describes the connectivity-occupancy correlation is presented without any explanation regarding the parameters that appear in this function: what do " ω ", " r " and " μ " describe? While " r " and " μ " are listed in Fig 1 (although one should at least add a sentence after the description of function $h^*(\omega)$ saying that these parameters are described in Fig 1), parameter " ω " is not mentioned anywhere. I assume that it is associated with the Fourier transform – but an ecology/biology student reading this manuscript will likely not figure out how " ω " appears here.

b. In the Optimal Foraging case, what does " $Poi(\lambda)$ " represents?

2) In Box 1 section E, the authors mention that a small parameter " ϵ " rescales the interaction kernels. It would be useful (especially for the students reading the manuscript) to add a sentence that would give more biological/ecological detail into the reasoning for this rescaling parameter.

3) Box 2: it is not very clear why the argument " k " (which has the same meaning in functions HPf_{ALL} and HGf_{ALL}) has different position in the two functions? What does the second argument " 1 " in HGf_{ALL} stand for?

4) Box 2: It may be useful to add a sentence in this box on how easy is to use a different type of kernel (i.e., not the top hat, but a Gaussian kernel, etc...)

5) Page 27: In the case studies sections in the Online Methods, it will be useful to remind the reader what is " h " (the disappearance rate of resources?) and how does " h " enter the integrand function $IntF(\omega)$ defined at the bottom of page 27

Responses to Reviewers:

Reviewers' comments are in *italics*; our responses are in upright font.

Reviewers' comments:

Reviewer #3 (Remarks to the Author):

This manuscript goes a long way to addressing concerns about the earlier version of the same paper. I enjoyed reading it, and also found the methods much clearer. I especially appreciate the inclusion of a worked example (Box 2), and the extent to which the revised manuscript tones down claims of being a "global" solution to all individual based models.

I am sure the work will be a valuable contribution to our understanding of spatial ecology and populatoin dynamics.

Now that I understand the manuscript better, I see that it solves one problem of individual-based simulation models, specifically (as the authors note) by providing numerical solutions rather than averaging over a number of replicate simulations, but does not solve another of their problems. The latter problem is the "black-box" nature of simulation models, which often have arbitrarily-specified parameters and hidden assumptions. Since we will rarely know the shape of many of the functions that go into these models, or the strength of the interactions among organisms, it would be easy to get results from this kind of model (or any simulation model) that depend mostly on the user's opinion, or desired output. I find that I prefer simpler models (simpler mathematically) because the assumptions are more transparent, and potentially more in line with the nature of ecological data. [This is more of a comment than a criticism.]

We thank the referee for their comments. We suggest, however, that our methods go some way to solving this problem because we are able to obtain analytical expressions rather than just numerical results. General conclusions can be drawn from these expressions (e.g. that the length scale of the optimal connectivity kernel is larger than that for the colonisation kernel) without making specific assumptions about parameters or kernel shapes.

This said, I am impressed by the work that has gone into this paper, and I am interested to see how the work gets used after it is published.

A few minor comments:

1st equation in Box 1: I really want the 2nd term on the RHS to be negative, even though it is technically correct to have it be positive

We think the Referee is suggesting that $D_m = q_m(\epsilon x, t) + \epsilon^d p_m(\epsilon x, t) + o(\epsilon^d)$ should be replaced by $D_m = q_m(\epsilon x, t) - \epsilon^d p_m(\epsilon x, t) + o(\epsilon^d)$, so that p_m would be positive if stochasticity reduces the density relative to the mean field. We note that either definition could be used, without loss of generality. However, we think our original notation is preferable because this is just one of many terms in a perturbation expansion and corrections to the mean field density can be positive rather than negative (for an example, see Fig. 6 in S. J. Cornell, O. Ovaskainen, *Theor. Popul. Biol.* 74, 209–225 (2008)). We feel it is likely to be less confusing if the sign of the corrections is the same as the sign of the quantity p_m .

Box 2, 2nd page: I do not understand the phrase "the name of the spatial frequency" - is this the name of a standard probability distribution or kernel used in the model?

On re-reading our MS, we recognize that the wording we used is prone to misinterpretation and could be improved. The term “spatial frequency” is used when discussing Fourier transforms of functions of both time and space; the conjugate variable to time is called “temporal frequency” and the conjugate variable to space is called “spatial frequency”. By “name of the spatial frequency” we mean “name of the variable that Mathematica will use to denote the conjugate variable to space”.

We have modified the wording to make it clearer, and – since we are only concerned with spatial Fourier transforms, so there is no need to distinguish between spatial and temporal frequencies – we now just used the term “frequency”.

I believe there is a typo in the legend for figure 2, where it says "length scale lambda of the connectivity kernel to the length scale l_c of the connectivity kernel" [should one "connectivity" be "dispersal"?)

We thank the referee for pointing out this error. Moreover, when correcting this, we noticed that we had used the two terms “dispersal kernel” and “colonisation kernel” interchangeably, so to avoid confusion we now use “colonisation kernel” throughout.

Reviewer #5 (Remarks to the Author):

The authors introduce a new framework for the analysis of various individual-based models. The approach is then applied to three case studies from movement ecology, evolutionary ecology and conservation biology.

The approach is very interesting and has the potential to change the field. Therefore the manuscript deserves publication.

However, before that, there are still a few aspects that still need to be addressed:

1) The Framework is presented briefly, and then applied to three different case studies, which are not always easy to understand (when reading the main manuscript).

a. For example, in the Optimal Landscape Connectivity case, the function $h^(\omega)$ that describes the connectivity-occupancy correlation is presented without any explanation regarding the parameters that appear in this function: what do “ ω ”, “ r ” and “ μ ” describe? While “ r ” and “ μ ” are listed in Fig 1 (although one should at least add a sentence after the description of function $h^*(\omega)$ saying that these parameters are described in Fig 1), parameter “ ω ” is not mentioned anywhere. I assume that it is associated with the Fourier transform – but an ecology/biology student reading this manuscript will likely not figure out how “ ω ” appears here.*

We thank the referee for pointing out this oversight in our original MS. The parameters r , μ , e are defined in the worked example, which was previously in “Box 2” and which now appears directly before this section. We now give definitions of these variables here as well as referring to the previous section. The referee is correct that the variable ω is the conjugate variable to space, that arises due to the Fourier Transforms. We now explain this in more detail.

b. In the Optimal Foraging case, what does “ $Poi(\lambda)$ ” represents?

We apologise, as we thought this would be a widely understood shorthand. We now state explicitly that this means a Poisson distributed random number with intensity λ .

2) In Box 1 section E, the authors mention that a small parameter “ ϵ ” rescales the interaction kernels. It would be useful (especially for the students reading the manuscript) to add a sentence that would give more biological/ecological detail into the reasoning for this rescaling parameter.

We have given a sentence explaining the significance of epsilon in Box 1, (which is now Figure 1) panel E, before equation (1). The ecological meaning of this is explained further in paragraph 2 of “The Framework”.

3) Box 2: it is not very clear why the argument “k” (which has the same meaning in functions HPfALL and HGfALL) has different position in the two functions? What does the second argument “1” in HGfALL stand for?

This was not very clear in our MS, and we have modified it accordingly. Unfortunately, these different functions need a different number of arguments and there is no sensible way to arrange them so that the arguments appear in the same place for both functions. The reasons for the differences in the arguments for the two functions are (i) HPfALL depends on one species label (because it relates to a population density) whereas HGfALL depends on two species labels (because it is a correlation function) (ii) The expression for HPfALL depends on spatial dimension (because it includes an integral over space), whereas the expression for HGfALL does not.

4) Box 2: It may be useful to add a sentence in this box on how easy is to use a different type of kernel (i.e., not the top hat, but a Gaussian kernel, etc...)

We now include, at the end of the worked example, instructions for how to use a Gaussian kernel within Model Simulator

5) Page 27: In the case studies sections in the Online Methods, it will be useful to remind the reader what is “h” (the disappearance rate of resources?) and how does “h” enter the integrant function $IntF(\omega)$ defined at the bottom of page 27

This parameter h is not a biologically interesting quantity, but instead an intermediate step in a convenient method for generating clusters of resources simultaneously. Resources are generated by a “target generator”, which disappears at rate h ; if the rate at which resources are generated is also proportional to h , then when we take the limit h approaches infinity the number of resources created by each target generator is finite. After taking this limit, the results no longer depend on h (which is infinite). This is now explained in more detail in section “Optimal foraging model” in the Online Methods.

Reviewers' Comments:

Reviewer #3:

None

Reviewer #5:

Remarks to the Author:

The authors have addressed appropriately the previous issues, and the manuscript can be published